# Investigation of An Extreme Rainfall Event during 8-12 December 2018 over Central Vietnam. Part I: Analysis and Cloud-Resolving Simulation

Chung-Chieh Wang and Duc Van Nguyen*

Department of Earth Sciences, National Taiwan Normal University, Taipei, Taiwan

Corresponding author address: Duc Van Nguyen (nguyenvanduc_t57@hus.edu.vn), Department of Earth Sciences, National Taiwan Normal University, No. 88, Sec. 4, Ting-Chou Rd., Taipei 11677, Taiwan

Highlights:

- A record-breaking rainfall event over central Vietnam is investigated
- Key factors in this event include the combined effect of northeasterly wind that originated from northern China, low-level easterly wind blow to central Vietnam from the northwest Pacific Ocean, southeasterly wind, local topography, and high sea surface temperature over North West Pacific Ocean and South China Sea.
- A cloud-resolving model is applied to simulated this extreme rainfall event in central Vietnam, and the results show that the model mostly captured the quantitative rainfall of this event. These results are very impressive.

**Abstract**

An extreme rainfall event occurred from 8 to 12 December 2018 along the coast of central
Vietnam. The observed maximum rainfall amount in 72 h was over 900 mm and set a new record,
and the associated heavy losses were also significant. The analysis of this event shows some key
factors for its occurrence: (1) The interaction between the strong northeasterly winds, blowing from
the Yellow Sea into the northern South China Sea (SCS), and easterly winds over the SCS in the
lower troposphere (below 700 hPa). This interaction created strong low-level convergence, as the
winds continued to blow into central Vietnam against the Truong Son Range, resulting in forced
uplift over the coastal plains due to the terrain's barrier effect. Furthermore, the low-level
convergence in this event was strong enough, and the air was unstable enough to trigger most of the
convection near the shoreline (further inland). As a consequence, heavy rainfall occurred along the
coastal zone and coastal sea. (2) The strong easterly wind played an important role in transporting
moisture from the western North Pacific across the Philippines and the SCS into central Vietnam.
(3) The Truong Son Range also contributed to this event due to its barrier effect. (4) In addition to
cumulonimbus, the low-level precipitating clouds such as nimbostratus clouds were also major
contributors to rainfall accumulation for the whole event. The analyses of local thermodynamics
also indicate that the southward movement of the low-level wind convergence zone caused the
southward movement of the main heavy rain band during the event.
The Cloud-Resolving Storm Simulator (CReSS) was employed to simulate this record-
breaking event at a grid size of 2.5 km, and evaluated results show the model had good simulated
the surface wind as well as captured the southward movement of the low-level wind convergence.
The overall rainfall can be captured quite well not only in quantity but also in its spatial distribution
(with a Fractions Skill Score $\approx 0.7$ and Threat Score $> 0$ at 700 mm for 72 h rainfall). Thus, the
CReSS model is shown to be a useful tool for both research and forecasts of heavy rainfall in
Vietnam. The model performed better for the rainfall during 9-10 but not as good on 11 December.
In the sensitivity test without the terrain, the model had poorly simulated the surface wind, which
led to the model not only did not generate nearly as much rainfall for this event but also did not
capture the spatial distribution of the rainfall. Thus, the test confirms the important role played by
the local topography for the occurrence of this event.
Keywords: Extreme rainfall, central Vietnam, cloud-resolving model.
**1 Introduction**
Heavy to extreme rainfalls are natural disasters that often cause deaths, flooding, landslides,
and erosion. Vietnam is one of the most disaster-prone countries in the world with many different
types of natural disasters. In the country, central Vietnam is most affected by natural disasters and
climate change, with frequent occurrences of rainstorms and extreme rainfalls. For example, during
8-12 December 2018, an extreme rainfall event (hereafter abbreviated as the D18 event) occurred
along the coast of central Vietnam. The peak 72-h accumulated rainfall (from 1200 UTC 8 to 1200
UTC 11 Dec) at some stations exceeds 800 mm (Fig. 1d). Among the stations, Da Nang (16.0° N,
108.2° E, cf. Figs. 1a,b) recorded 24-h rainfall amounts greater than 600 mm on 9 December and
over 300 mm the next day. This extreme event resulted in 13 deaths, an estimated 1200 houses
inundated, around 12,000 hectares of crops destroyed, some 160,000 livestock killed and many
other economic losses (Tuoi Tre news, 2018). Furthermore, according to a publication by the
Ministry of Natural Resources and Environment of Vietnam (Tran *et al.*, 2016) regarding climate
change and sea-level rise scenarios, extreme precipitation events will increase in both their
frequency and intensity in the future. Hence, how to improve the ability in the quantitative
precipitation forecast (QPF) of heavy-rainfall events over central Vietnam is very important.
Climatologically, the central part of Vietnam is the country's rainiest region and is strongly
affected by heavy to extreme rainfall, with average annual precipitation ranging from 2400 to over
3300 mm (1980–2010, Fig. 1f). The main rainy season in this region is from late fall to early winter
(Yokoi and Matsumoto, 2008; Chen *et al.*, 2012). Past studies have shown some main factors that
can lead to heavy rainfall in central Vietnam, such as (1) the combined effect of cold surges that
originate from northern China, (2) tropical depressions, and (3) local topography due to the
topography is characterized by high mountains (< 3000 m), highlands, narrow coastal plain with the
narrowest place less than 100 km in width (east-west), and gradually lowers from the west to the
east (Fig. 1a) (Bui, 2019; Yokoi and Matsumoto, 2008; Chen *et al.*, 2012; Nguyen-Le and
Matsumoto, 2016; van der Linden *et al.*, 2016). According to these studies, a cool, dry continental
surface high pressure system (known as the Siberian high-pressure system) gradually establishes
over the continental East Asia after boreal summer in October–November. This high-pressure
system's intensification and southeastward amplification lead to an episodic southward progression
of cold surge into the tropics. The interaction of this cold surge and preexisting tropical disturbance
over the SCS and the topography in central Vietnam can bring large amounts of rainfall along the
east-central coast through orographic rainfall processes.

In this study, central Vietnam is referred to as the area between 14.7° N and 18° N (Fig. 2a). Its

eastern boundary is the South China Sea (SCS), and the western boundary is the border to Laos,
where the Truong Son Range (also known as the Annamite Range) runs parallel to the coast. The
central Vietnam includes Quang Binh, Quang Tri, Thua Thien Hue, Da Nang city, Quang Nam, and
a part of Quang Ngai province. Most of the population and cities are concentrated along the coastal
plain. By these characteristics of steep topography, when heavy rain occurs, it often leads to
flooding and causes great damages to people and the environment.




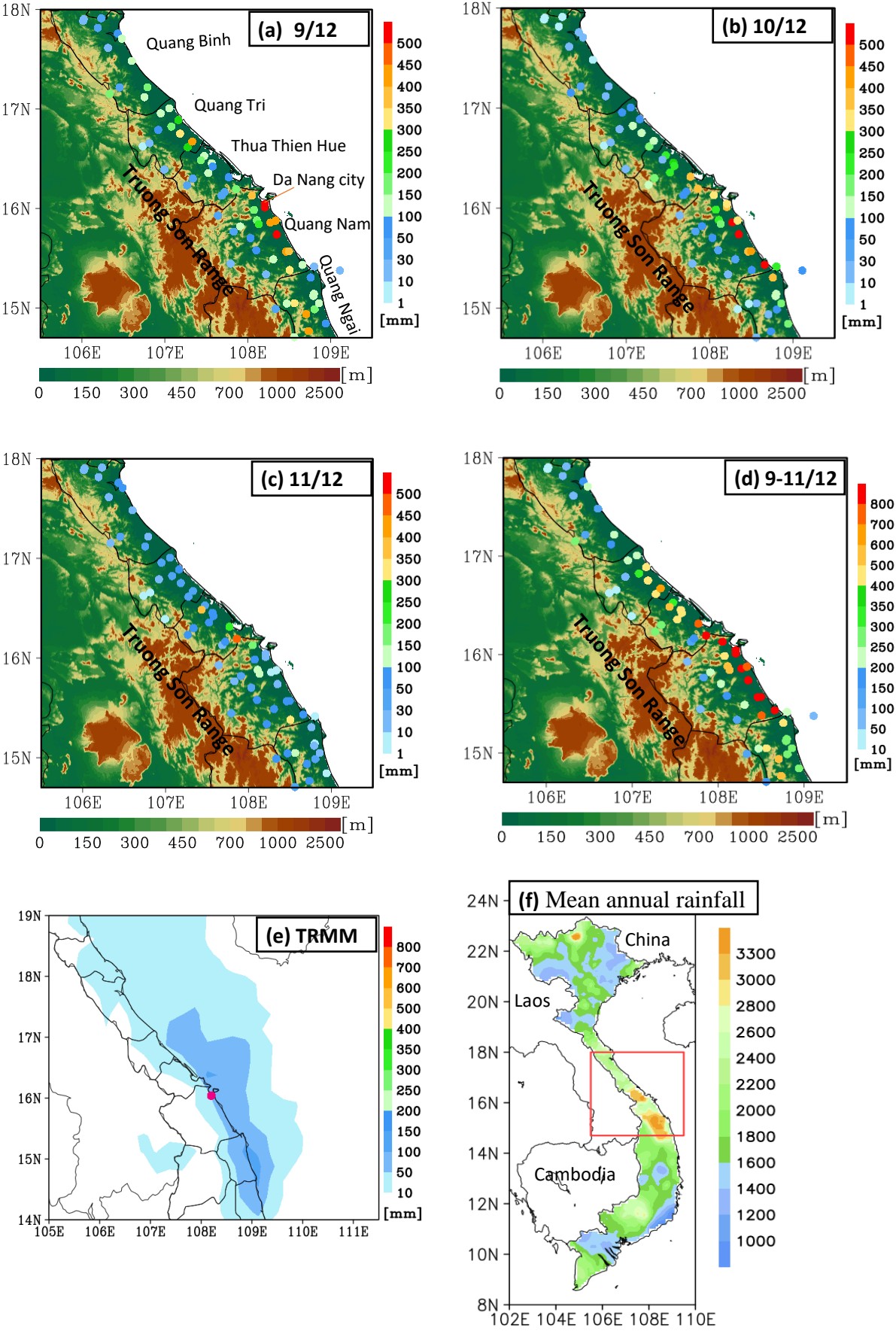

**Figure 1.** (a) observed 24 h accumulated rainfall (mm, color dots, 1200 – 1200 UTC) and topography (m, shaded) for 9 Dec. Vertical colorbar for rainfall, and horizontal colorbar for topography. (b) As in (a), but for 10 Dec. (c) As in (a), but for 11 Dec. (d) As in (a), but for 72 h accumulated rainfall during 1200 UTC 8–1200 UTC 11 Dec. (e) 72 h accumulated rainfall obtained by TRMM estimate. The pink dot marks the location of Da Nang station. (f) Mean annual rainfall distribution (mm) in Vietnam from 1980 to 2010, obtained from the Vietnam Gridded Precipitation (VnGP) data, and the study area of central Vietnam (red box).

Furthermore, according to Wang et al. (2017), Vietnam is impacted by about 4-6 typhoons per year. Nguyen-Thi *et al.* (2012) investigated the characteristic of tropical cyclone rainfall over Vietnam in the climatology. Their results show that the tropical cyclone rainfall amount is concentrated in central Vietnam, peaking between October and November. Takahashi et al. (2009) performed a long-term simulation for September (from 1966 to 1995) using a high-resolution model. They found that the observed long-term decrease in September rainfall is due to the weakening of tropical cyclone activity over the Indochina Peninsula. As for the impacts of El Niño-Southern Oscillation (ENSO), some studies have examined the linkages between rainfall in Vietnam and ENSO, and suggested more (less) rainfall during La Niña (El Niño) years. For example, Yen *et al.* (2010) analyzed the interannual variation of the rainfall in fall over central Vietnam, and their results indicated a negatively correlated relationship between rainfall in central Vietnam and the sea surface temperature over the NINO3.4 region. Besides, Vu et al. (2015) investigated the effects of ENSO on fall rainfall in central Vietnam and concluded that central Vietnam has more (less) rainfall in La Niña (El Niño) years. Finally, Wu et al. (2012) analyzed the Madden-Julian Oscillation (MJO) activity from September to November for 30 years (1981-2010) over Vietnam and showed that the MJO is also an important factor in the formation of extreme precipitation events in central Vietnam.

In recent decades, the Cloud-Resolving Storm Simulator (CReSS) has been widely known due to its good performance in quantitative precipitation forecasts. This model has been applied to study tropical cyclones, heavy to extreme rainfall events, and many other convective systems in Japan and

Taiwan (e.g., Ohigashi and Tsuboki, 2007; Yamada *et al.*, 2007; Akter and Tsuboki, 2010, 2012;
Wang *et al.*, 2015). Furthermore, the CReSS model has been used to perform routine high-
resolution forecasts at the National Taiwan Normal University (NTNU) and provided to the TTFRI
as a forecast member since 2010. Hence, this study employed the CReSS model to simulate the
D18 event and evaluated its performance
From the review above, the important mechanisms for the heavy rainfall in some previous
events over central Vietnam are revealed. However, according to Dr. Hoang Phuc Lam – National
Center for Hydro- Meteorological Forecasting, it can be said that this extreme event has never
happened in the past because the observed rainfall at some places in the Central region has
surpassed the record according to the statistics of rainfall at the end of the main rainy season
(Communist Party of Vietnam Online Newspaper). Furthermore, Figs.1a,b,c,d and e show that the
main heavy rain band concentrated on the coastal plains and coastal sea while Fig. 1f show the
annual mean rainfall extends into the mountain with their peak amounts over the mountain slopes,
several questions are therefore raised: What mechanisms caused this record-breaking event at such
a magnitude? Was its mechanism similar to those in previous events? Or, it was a different one.
How important was the role played by local terrain in this event? From a forecast perspective, one
related question would be whether a cloud-resolving model is capable of reproducing the D18
event? The answers to these questions will help improve our understanding on the mechanisms that
cause heavy rainfall in central Vietnam, as well as on the predictability of such events in the future.
Hence, the present study was carried out with an aim to answer the above questions. The remainder
of this paper is organized as follows: Section 2 describes the datasets and methodology used in the
study. The analysis and modeling results are presented in Section 3 and 4, respectively. Finally, the
conclusions are given in Section 5.
**2 Data and Methodology**
**2.1 Data**
*2.1.1 NCEP GDAS/FNL Global Gridded Analyses and Forecasts*
The NCEP GDAS/FNL Global Gridded Analyses and Forecasts is provided freely by the
National Centers for Environmental Prediction (NCEP). In this study, this dataset is used as the
initial and boundary conditions (IC/BCs) for the cloud-resolving model (CRM) simulation. The data
are on a $0.25° \times 0.25°$ latitude-longitude grid with 26 levels extending from the surface to 20 hPa.
The data period is from 0600 UTC 8 December to 0000 UTC 13 December 2018, at 6-h intervals.
Parameters include geopotential height, zonal and meridional wind components, pressure,
temperature, and relative humidity. The dataset and its detailed information are available at
https://rda.ucar.edu/datasets/ds083.3.
*2.1.2 The fifth generation ECMWF reanalysis data (ERA5)*
The ERA5 is the fifth-generation reanalysis dataset, developed by the European Centre for
Medium-range Weather Forecasts (ECMWF) to replaces the ERA-Interim reanalysis. We have used
these data to delineate the synoptic weather patterns during the D18 event. The horizontal resolution
of this dataset is $0.25° \times 0.25°$ latitude-longitude at 22 selected levels from 1000 to 100 hPa and
including the surface. Parameters include zonal and meridional wind components, geopotential
height, specific humidity, relative humidity, temperature, vertical velocity, mean sea level pressure,
and sea surface temperature. The dataset was downloaded from 0000 UTC 8 to 1800 UTC 11
December 2018 at 6-h intervals (Hersbach et al., 2018a,b).
*2.1.3 Observation data*
The daily observed rainfall data (1200–1200 UTC, i.e., 1900–1900 LST) from 8 to 12
December 2018 at 69 automated gauge stations across central Vietnam are used for case overview
and verification of model results. This dataset is provided by the Mid-central Regional Hydro-
Meteorological Centre, Vietnam.
*2.1.4 Satellite data*

(a) TRMM (TMPA) rainfall estimates

The TRMM multi-satellite precipitation estimates (3B42, version 7, Huffman *et al.*, 2016) are

freely provided by the NASA Goddard Earth Sciences (GES) Data and Information Services Center
(DISC). The horizontal resolution of this dataset (level 3) is $0.25° \times 0.25°$ latitude-longitude and the
time resolution is every 3 h. In this study, we used this satellite data to verify rainfall distribution
over the coastal sea due to the limitation of the observation station network, we only have the
observation stations inland, as shown in the Figure. 1d and Fig. 1e. This dataset was downloaded
from 1200 UTC 8 to 1200 UTC 11 December 2018 to analyze the D18 event.

(b) The Himawari satellite images

The color-enhanced infrared imageries are designed mainly for the detection of convective

clouds, including those from the Himawari-8 satellite. The different colours represent different
cloud-top heights. Therefore, we have used these images to discern deep convection in convective
clouds and precipitating clouds based on their characteristics. In this study, the dataset was
downloaded from the Central Weather Bureau website, Taiwan, with a time resolution of 1 h.
*2.1.5 Radar data*

The column-maximum radar reflectivity data are one indispensable data source to identify

precipitation and verify model results. The reflectivity data (in dBZ) cover a wide range and the
values indicate rainfall intensity (the higher the dBZ, the stronger the intensity of precipitation).
Therefore, we used the column-maximum radar reflectivity data over central Vietnam at 1-h
intervals over 8-11 December 2018 to estimate the rainfall intensity during the D18 event. This
dataset is provided by the Mid-central Regional Hydro-Meteorological Centre of Vietnam.
*2.1.6 The Vietnam Gridded Precipitation (VnGP) Dataset.*

42

The VnGP data are derived base on the daily observed data from 481 rain gauges cross
Vietnam. This dataset has a resolution of 0.1° and covers the period of 1980-2010 (Nguyen-Xuan et
al., 2016). In this study, this dataset is used to depict the rainfall climatology in Vietnam.
*2.1.7 The Oceanic Niño Index (ONI) data*
The Oceanic Niño Index (ONI) data was made and provided freely by NOAA Climate
Prediction Center (CPC). The ONI data was computed by three month running mean of NOAA
ERSST.V5 SST anomalies in the Niño 3.4 region (5N-5S, 120-170W), based on changing base
period which onsist of multiple centered 30-year base periods. The ONI is the most commonly used
indices to define El Niño and La Niña events. This study used the ONI data for Niño 3.4 region to
define the ENSO phase of 2018. This data is available at:
https://psl.noaa.gov/data/correlation/oni.data
**2.2 Model description and experiment setup**
The Cloud Resolving Storm Simulator (CReSS, version 3.4.2), developed by Nagoya
University, Japan (Tsuboki and Sakakibara, 2002, 2007) is used for numerical simulation of the
D18 event. This model is a non-hydrostatic and compressible cloud model, designed for simulation
of weather events at high (cloud-resolving) resolution. In the model, the cloud microphysics is
treated explicitly at the user-selected degree of complexity, such as the bulk cold-rain scheme with
six species: vapor, cloud water, cloud ice, rain, snow, and graupel (Lin *et al.*, 1983; Cotton *et al.*,
1986; Murakami, 1990, 1994; Ikawa and Saito, 1991). Other subgrid-scale processes parameterized,
such as turbulent mixing in the planetary boundary layer, as well as physical options for surface
processes, including momentum/energy fluxes, shortwave and longwave radiation are summarized
in Table 1.
To study the D18 event and investigate the role played by the local terrain in this event using
the CReSS model, two experiments were performed using the same model domain setting, physical
options, and initial and boundary conditions. Specifically, both experiments using a single domain
at 2.5-km horizontal grid spacing and a (x, y, z) dimension of 912 x 900 x 60 grid points (Table 1,
cf. Figure 2). As introduced in subsection 2.1.1, the NCEP GDAS/FNL Global Gridded Analyses
and Forecasts (0.25° x 0.25°, every 6 h, 26 pressure levels) was used as the IC/BCs of the model.
These experiments were started from 0600 UTC 8 to 0000 UTC 13 December 2018 (for a
simulation length of 114 h).
The only different setting between these experiments is at the lower boundary, the real terrain
data at (1/120∘) resolution (roughly 0.9 km) was provided for the control simulation (CTRL) while
this was ignored for the sensitivity test without the terrain (NTRN)
The main information of these two experiments, including the domain setup and model
configuration, is listed in Table 1.

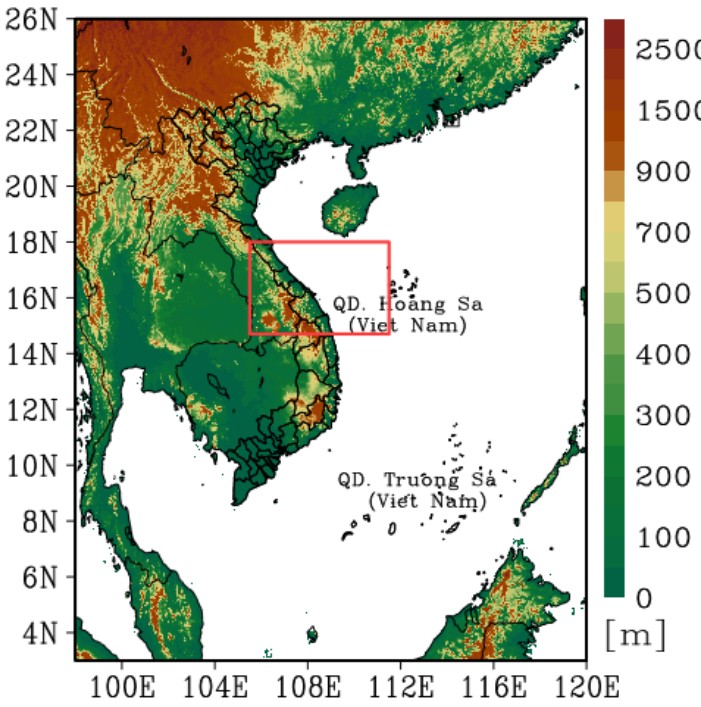


**Figure 2**: The simulation domain of the CReSS model and topography (m) used in this study. The
red box marks the study area.
**Table 1.** The basic information of experiments.

| Domain and Basic setup | |
|---|---|
| Model domain | 3°–26°N; 98°–120°E |
| Grid dimension ($x$, $y$, $z$) | $912 \times 900 \times 60$ |
| Grid spacing ($x$, $y$, $z$) | 2.5 km $\times$ 2.5 km $\times$ 0.5 km* |
| Projection | Mercator |
| IC/BCs (including SST) | *NCEP GDAS/FNL Global Gridded Analyses and Forecasts (*$0.25° \times 0.25°$*, every 6 h, 26 pressure levels)* |
| Topography (for CTRL only) | Digital elevation model by JMA at $(1/120)°$ spatial resolution |
| Simulation length | 114 h |
| Output frequency | 1 hour |
| Model physical setup | |
| Cloud microphysics | Bulk cold-rain scheme (six species) |
| PBL parameterization | 1.5-order closure with prediction of turbulent kinetic energy (Deardorff, 1980; Tsuboki and Sakakibara, 2007) |
| Surface processes | Energy and momentum fluxes, shortwave and longwave radiation (Kondo, 1976; Louis et al., 1982; Segami et al., 1989) |
| Soil model | 41 levels, every 5 cm deep to 2 m |

* The vertical grid spacing (Δz) of CReSS is stretched (smallest at bottom) and the averaged value is
given in the parentheses
**2.3 Verification of model rainfall**

In order to verify the model-simulated rainfall, some verification methods are used, including

(1) visual comparison between the model and the observation (from the 69 automated gauges over
the study area), and (2) the objective verification using categorical skill scores at various rainfall
thresholds from the lowest at 0.05 mm up to 900 mm for three-day total. These scores are listed in
Table 2 along with their formulas, perfect value, and worst value, respectively. To apply these
scores at a given threshold, the model and observed value pairs at all verification points (gauge sites
here, N) are first compared and classified to construct a $2 \times 2$ contingency table (Wilks, 2006). At
any given site, if the event takes place (reaching the threshold) in both model and observation, the
prediction is considered a hit (H). If the event occurs only in observation but not the model, it is a
miss (M). If the event is predicted in the model but not observed, it is a false alarm (FA). Finally, if
both model and observation show no event, the outcome is correct rejection (CR). After all the
points are classified into the above four categories, the scores can be calculated by their
corresponding formula in Table2.
**Table 2.** List of the categorical skill scores and their formulas.

| Name of skill score | Formula | Perfect score | Worst score |
|---|---|---|---|
| Bias Score (BS) | (H+FA)/(H+M) | 1 | 0 or N - 1 |
| Probability of Detection (POD) | H/(H+M) | 1 | 0 |
| False Alarms Ratio (FAR) | FA/(H+FA) | 0 | 1 |
| Threat Score (TS) | H/(H+M+FA) | 1 | 0 |


In addition to the categorical scores, the Similarity Skill Score (SSS, Wang et al., 2022) is also
applied to evaluate the model rainfall results, as
$$\text{SSS} = 1 - \frac{\frac{1}{N}\sum_{i=1}^{N}(F_i - O_i)^2}{\frac{1}{N}\sum_{i=1}^{N}F_i^2 + \frac{1}{N}\sum_{i=1}^{N}O_i^2} \tag{1}$$
where N is the total number of verification points, $F_i$ is the forecast value, and $O_i$ is the observed
value, at the *i*th point among N, respectively. SSS is used to measure against the worst the mean
squared error (MSE) possible. The formula shows that a forecast with perfect skill has a FSS of 1,
while a score of 0 means zero skill.
**3 Overview of the D18 Event**
**3.1 Rainfall and its distribution**
The maximum accumulated rainfall was recorded from 9 to 11 December with a peak daily
rainfall greater than 500 mm and 72-h accumulated rainfall exceeds 800 mm (Figs. 1a-d). Besides,
the daily and 72-h rainfalls observed at 69 stations show that the extreme precipitation occurred along
the eastern coastal plains, on the eastern side of the Truong Son Range. Especially over Quang Nam
province, where the Truong Son Range reaches its highest of over 2500 m (Figs. 1a-d). In addition,
satellite products from the Tropical Rainfall Measuring Mission (TRMM) seriously underestimates
the D18 event (Fig. 1e), but indicates that the rainfall occurred not only in coastal plains but also over
the nearby ocean.
**3.2 Synoptic conditions**

During the D18 event, the horizontal winds at 925 hPa (averaged from 0000 UTC 8 to 1800

UTC 11 December) over central Vietnam and the SCS are characterized by a strong convergent
zone between the northeasterly winds blowing from northeastern China into northern SCS and
central Vietnam, and the easterly winds blowing from the western North Pacific (WNP) into the
SCS (Fig. 3a). The wind speed over northern SCS and central Vietnam is over 13 m s$^{-1}$. At 850
hPa, horizontal winds are predominantly easterly, with speeds of about 10–13 m s$^{-1}$ (Fig. 3b). At
500 hPa, central Vietnam is affected by southeasterly winds that originated from the easterly winds
over the WNP (Fig. 3c). Besides, Figure 3 also indicates that there was no existence of any tropical
cyclone during the D18 event. Therefore, tropical cyclones or the combined effect of cold surges
originating from northern China and tropical depressions that have been mentioned as one of the
patterns that cause heavy rainfall in central Vietnam is not the mechanism of the D18 event.

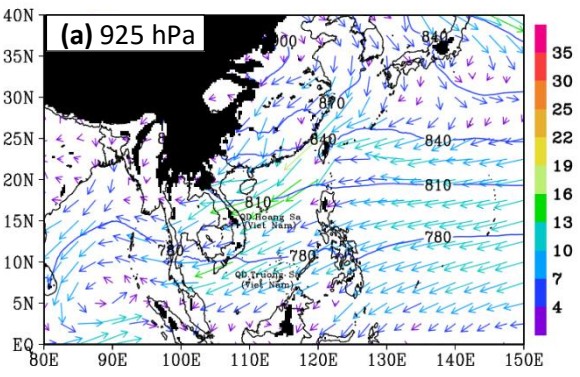 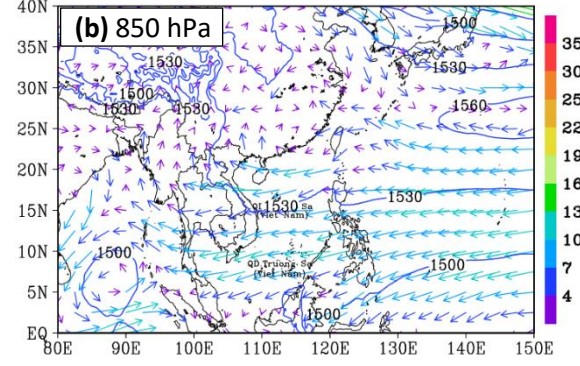

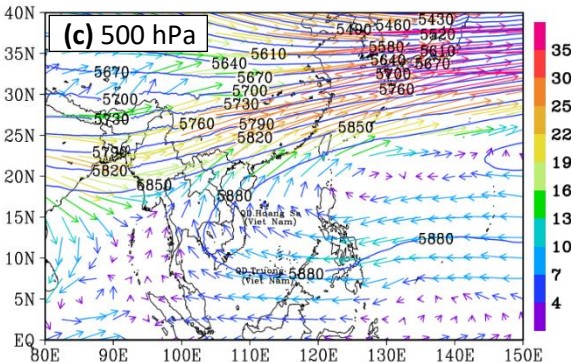

**Figure 3.** (a) The ERA5 averaged horizontal wind vectors (m s$^{-1}$, color for speed) and geopotential height (gpm, blue contours, every 30 gpm) at the 925 hPa from 0000 UTC 8 to 1800 UTC 11 Dec 2018. (b) As in (a), but for the 850 hPa. (c) As in (a), but for the 500 hPa. The blacked areas are where the 925-hPa level is below the ground.

From a thermodynamic perspective, the equivalent potential temperature ($\theta_e$) field at 925 hPa shows that a warm and moist tropical air mass exist in central and SCS with $\theta_e$ values greater than 335 K, and the relative humidity is around 90 % during the D18 event (Fig. 4a). The high moisture content combines with a decrease in $\theta_e$ with altitude, indicating convective instability in the lower atmosphere below about 500 hPa (Fig. 4b). Furthermore, the interaction between northeasterly and easterly winds seemed to enhance instability in the lower atmosphere.

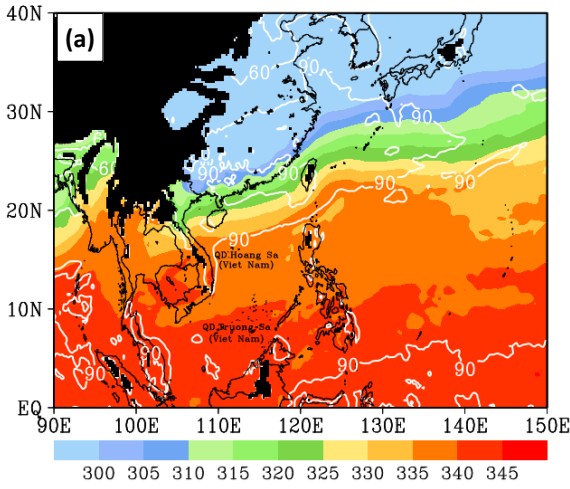
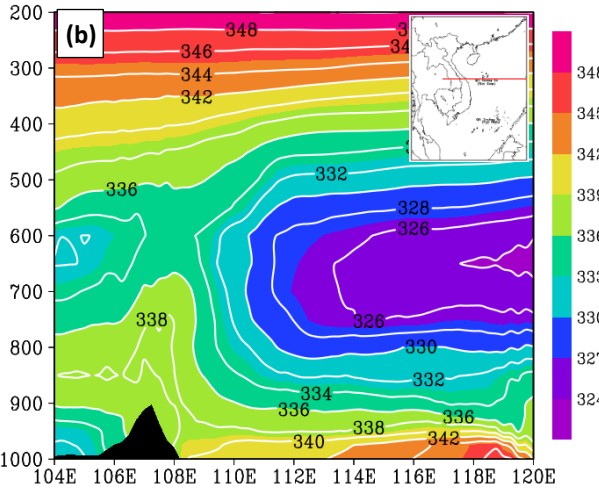

**Figure 4.** (a) The ERA5 averaged equivalent potential temperature (K, color), and relative humidity
(%, white contours, every 30 %) at 925 hPa. The blacked areas are where the 925-hPa level is
below the ground. (b) the east-west vertical cross-section along 16°N (see insert) of averaged
equivalent potential temperature ($\theta_e$, K, color, every 5 K), from 0000 UTC 8 to 1800 UTC 11 Dec
2018. The topography is dark shaded.
The above analysis suggests that the northeasterly, easterly, and southeasterly winds (cf. Figs.
3a-c) all played an important role in transported unstable air into central Vietnam. Particularly,
when the strong northeasterly and the easterly winds at low levels and southeasterly wind at upper
levels blow into central Vietnam, they bring warm, moist, and unstable air into central Vietnam.
This moisture is transported to central Vietnam by strong moisture flux through the deep column
from the WNP, across the Philippines and the SCS (Fig. 5a). Furthermore, the high SST of the SCS
(>27° C) also help to enhance and maintain abundant moisture during this event (Fig. 5b).

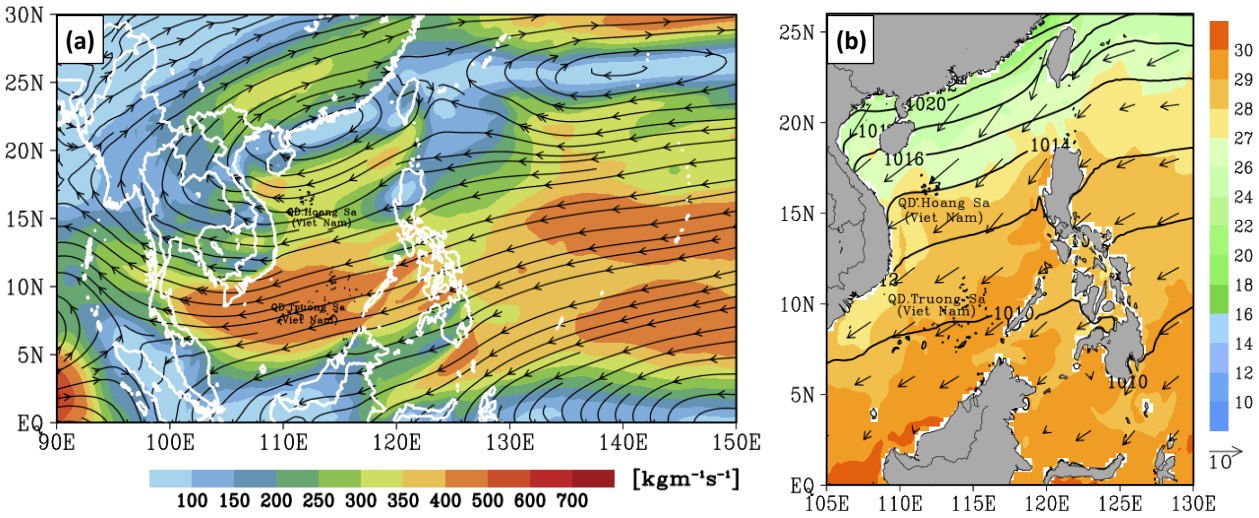

**Figure 5**. (a) The ERA5 averaged surface–200-hPa vertically integrated moisture flux (kg m$^{-1}$s$^{-1}$).
(b) the ERA5 averaged SST (°C, color), mean sea-level pressure (hPa, isobars, every 2 hPa), and
horizontal wind vectors at 10-m height (m s$^{-1}$, vector), from 0000 UTC 8 to 1800 UTC 11 Dec

2018.

Consequently, the atmospheric conditions and local topographic characteristics in interaction
result in moisture convergence and forced uplift in the lower troposphere during the D18 event.
This can be seen in Fig. 6, where extensive rising motion occurs in the lower troposphere along
coastal Vietnam, with a maximum value of -1.2 Pa s$^{-1}$. Besides, Figs. 6a,b also indicate that the
strong northeasterly wind along with warm, moist and unstable air is blocked by the Truong Son
Range. This pattern suggests that the Truong Son Range also played an important role in the
development of heavy rainfall in central Vietnam in D18. In detail, when the northeasterly and
easterly winds at low levels blow into central Vietnam and become block by the Truong Son Range,
which is located along the border of Vietnam and Laos, forced uplift is resulted at the windward
side, with downward motion over the lee side (in Laos, Fig. 6b). Furthermore, the low-level
convergence in this event was strong enough (Fig. 3a), and the air was unstable enough (Fig. 4b) to
trigger most of the convection near the shoreline (further inland, Fig. 6a)

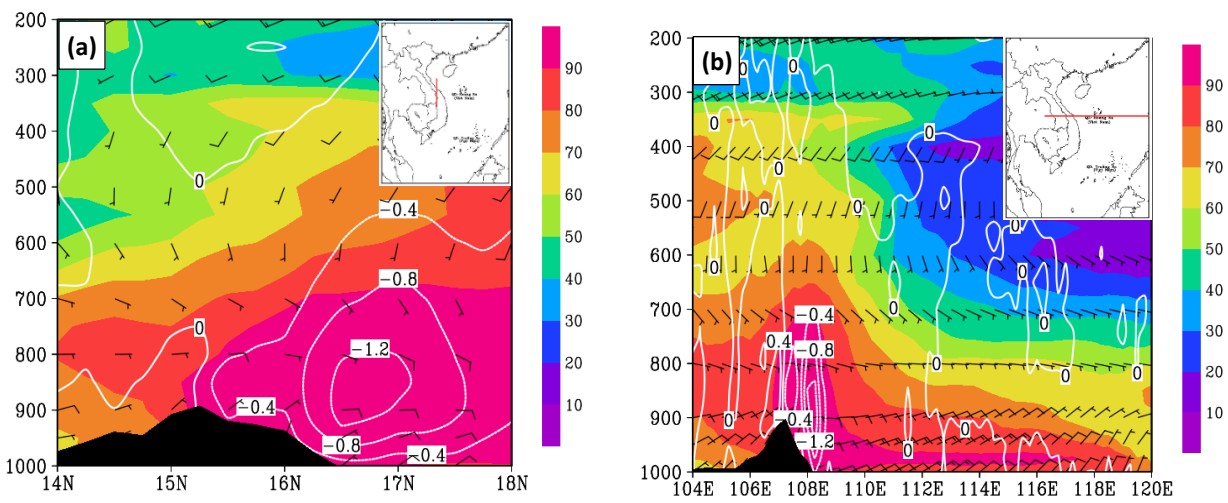

**Figure 6.** (a) The ERA5 the south-north vertical cross-section along 107.5°E (see insert) of
averaged horizontal wind (m s$^{-1}$, vectors) and vertical motions (Pa s$^{-1}$; white contours, negative for
upward motion), and relative humidity (%, shaded), from 0000 UTC 8 to 1800 UTC 11 Dec 2018.
The topography is dark shaded. (b) As in (a), but for the vertical cross-section along 16° N.
As described above, when the strong northeasterly and easterly winds at low levels blow into
central Vietnam, they bring warm, moist, and unstable air that originated in the WNP and is
enhanced over the SCS. Then, this air is blocked by the Truong Son Range, which has a height of
around 2 km, leading to forced convergence and upward motion at low levels and divergence
further above. These conditions consequently lead to moisture flux convergence of over $8 \times 10^{-4}$ g
$kg^{-1} s^{-1}$ at 925 hPa (Fig. 7a) and moisture flux divergence at 850 hPa with comparable magnitudes
(Fig. 7b). This divergence reduces sharply further up toward the middle and upper levels (Fig. 7c).
These factors create a moist atmosphere with a precipitable water amount (through the deep
column) exceeding 50 mm during the D18 event (Fig. 7d). The above atmospheric ingredients and
characteristics in local topography in combination created favorable environmental conditions to
trigger orographic rainfall. As a consequence, the D18 event happened.

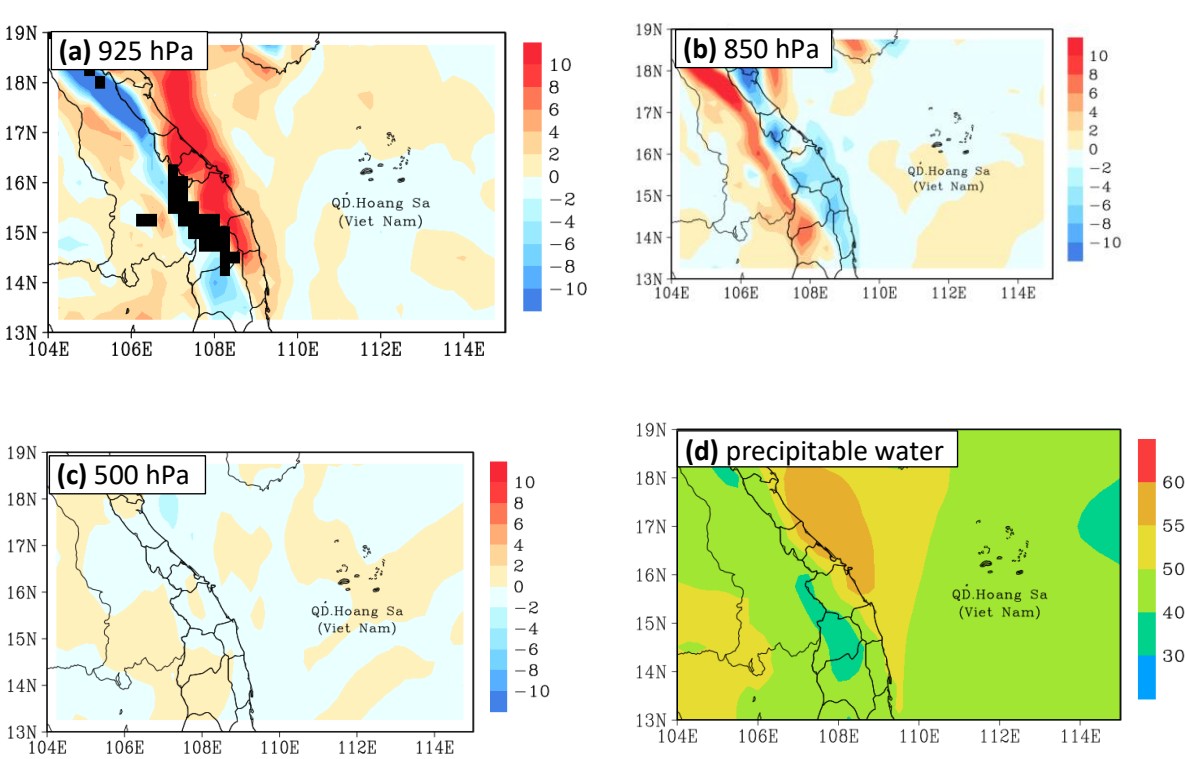

**Figure 7.** (a) The ERA5 averaged moisture convergence/ divergence ($x10^{-4}$, g $kg^{-1} s^{-1}$, shaded,
positive for convergence) at the 925 hPa, from 0000 UTC 8 to 1800 UTC 11 Dec 2018. The blacked
areas are where the 925-hPa level is below the ground. (b) As in (a), but for the 850 hPa. (c) As in
(a), but for the 500 hPa. (d) The ERA5 averaged precipitable water between surface and 200 hPa
(mm), from 0000 UTC 8 to 1800 UTC 11 Dec 2018.

Besides investigating the synoptic-scale atmospheric conditions above, this study also verified

the impact of intraseasonal oscillations in the tropical atmosphere on the D18 event. To be more
specific, figure 8a reveals that the MJO in Western Pacific was not active in early December 2018
as well as during the D18 event. Figure 8b indicates that the last three months of 2018 are a fairly
weak El Niño phase. In addition, previous studies showed that central Vietnam had less rainfall in
the El Niño years. Therefore, MJO and ENSO are also not the cause and have no impact on the D18
event.

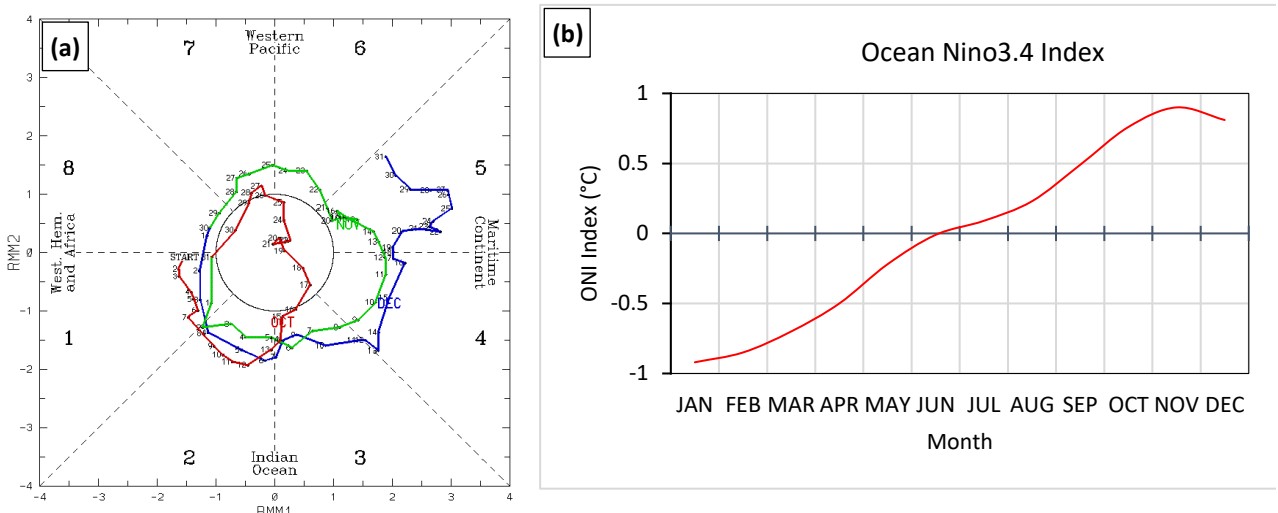

**Figure 8.** (a) The Madden-Julian Oscillation (MJO) location and the strength through 8 different
areas along the equator around the globe. Labelled dots for each day. Red line is for October, Green
line is for November, Blue line is for December. Source: Commonwealth of Australia 2019, Bureau
of Meteorology. (b) The Oceanic Niño Index (ONI) of the Niño 3.4 region (5° N-5° S, 120°-170°
W) for 2018.
**3.3 The local thermodynamic conditions prior the D18 event**

351   Figure 9 shows these conditions at 1200 UTC 8 December 2018. At this time, there is a strong

352 convergence zone of the low-level northeasterly wind carrying the moisture over the north of the

353 study area and near the shoreline (Figs. 9a,b). The northeasterly wind convergence led to a low-level

354 moisture convergence both inland and over the coastal sea. This happened as the low-level

355 northeasterly wind carrying the moisture blew to central Vietnam and interacted with local

356 topography, the low-level northeasterly flow reduced in speed over a wide area (refers to Figs. 6),

357 leading to a strong moisture flux convergence at low-level both inland and near the shoreline and

358 moisture flux divergence at the upper level (Figs. 9c, d). Due to the convergence of northeasterly

359 wind and moisture happened mainly in the north of latitude 16, the rising motion in the south of

360 latitude 16 mainly happened at low-level (less than 700 hPa, Fig. 9e) due to blocked by the Truong

361 Son range. Furthermore, this process occurred in a warm and unstable atmosphere (refer to Figs. 4),

362 making a favourable environmental condition to trigger most of the convection near the shoreline

363 instead of over the slopes (further inland) by forced uplift of the terrain. Hence, precipitable water

364 between the surface and 200 hPa exceeding 55 mm just formed over the coastal zone of the north of

365 the study area (Fig. 9f). Consequently, heavy rainfall only concentrated around the coastal zone.

366 These analyses are suitable for satellite and radar data.

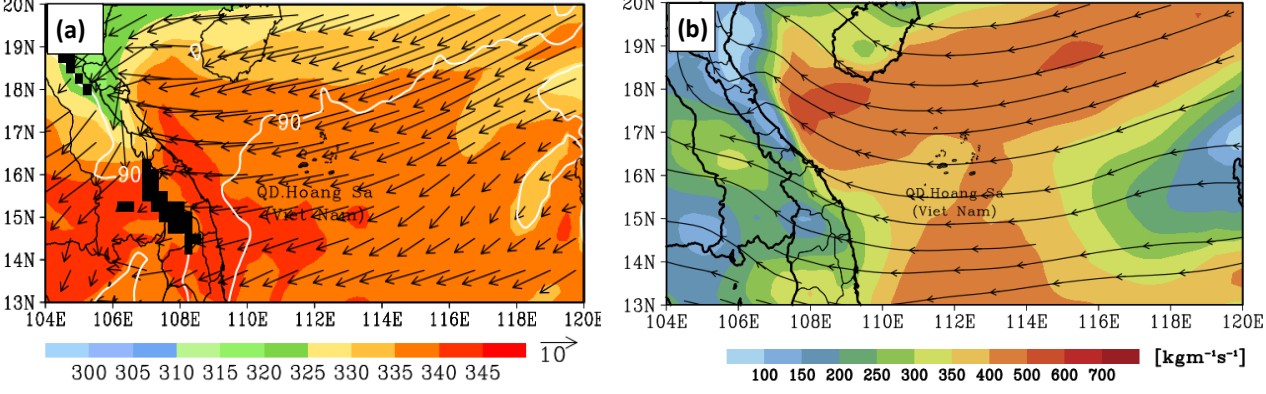

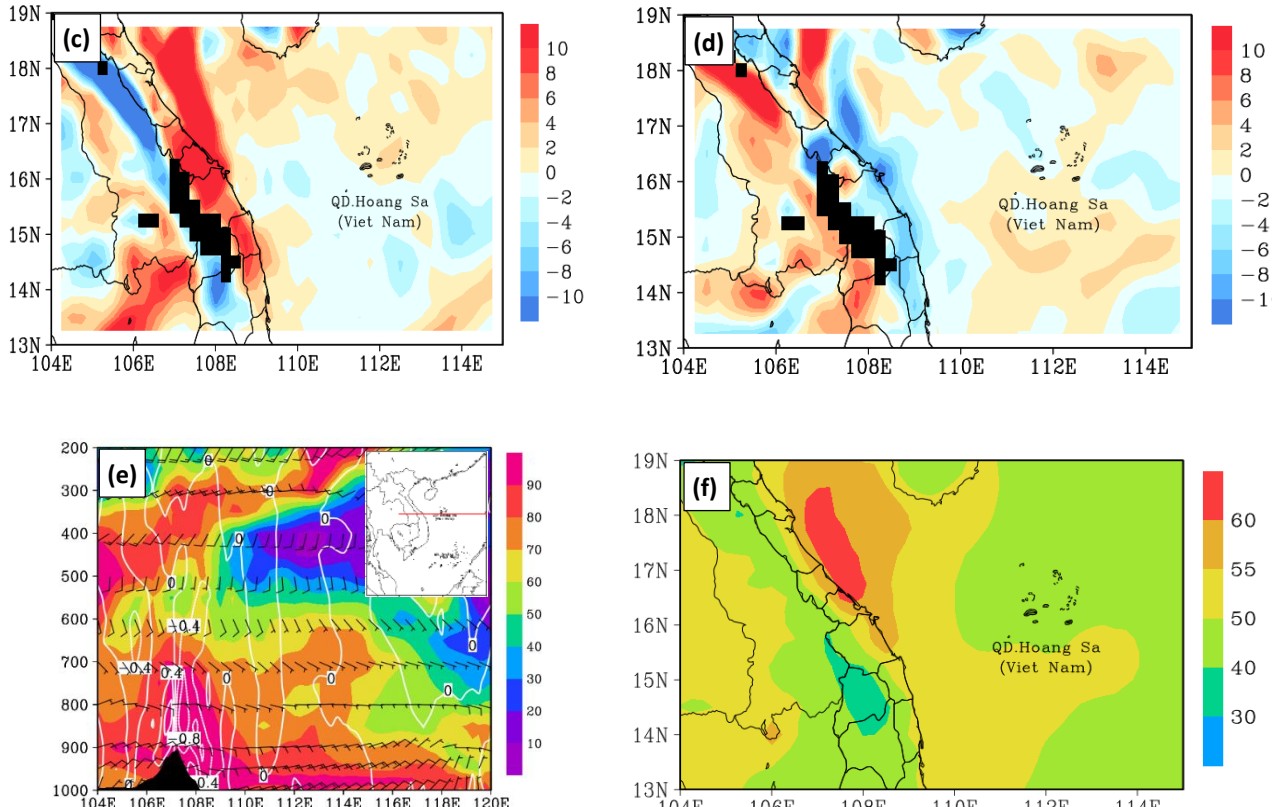

**Figure 9.** (a) The ERA5 $\theta_e$ (K, shaded), horizontal winds (m s$^{-1}$, vector), and relative humidity (%, white contours, every 30 gpm) at 925 hPa. The blacked areas are where the 925-hPa level is below the ground. (b) Surface–200-hPa vertically integrated moisture flux (kg m$^{-1}$ s$^{-1}$). (c) East-west vertical cross-section along 16°N (see insert) of vertical motions (Pa s$^{-1}$, white contours), relative humidity (%, shaded), and horizontal winds (m s$^{-1}$, vector). The topography is black shaded. (d) Precipitable water between surface and 200 hPa (mm). All panels are for 1200 UTC 8 Dec 2018.

To be more specific, on satellite imageries from 1200 UTC 8 to 1100 UTC 9 December (Fig. S1), a series of deep convective clouds (cumulonimbi, or Cb) first form over northern and central Vietnam and Laos on 8 December, with mainly a northeast-southwest to east-west alignment. With blackbody temperatures ($T_B$) below −42° C, several isolated deep cells also develop near the coast over the southern part of the study area after 0200 UTC on 9 December (Fig. S1). Generally, these deep Cb clouds tend to move slowly offshore and weaken after a few hours. Meanwhile, the study area is also covered by precipitating clouds known as nimbostratus (Ns) that are not as deep, with

cloud-top $T_B$ at $-20°$-$0°$ C and above (Fig. S1). These Ns clouds first form over the northern part of

the study area and then grow and expand southward along the coast, eventually cover the entire

study area on 9 December (Fig. S1). As analyzed above, both deep Cb clouds and the persistent Ns

clouds produced long-lasting rainfall for hours, starting along the coast from 1200 to 1700 UTC 8

December. After that, the rain area extends both inland and over the coastal sea (Fig. S2). The

rainfall intensity is the greatest from 2000 UTC 8 to 0200 UTC 9 December, with a column-

maximum radar reflectivity ($C_{max}$) $\approx$ 40 dBZ (Fig. S2). Afterwards, the rainfall intensity decreases

to some extent but remain at 15-35 dBZ rather steadily (Fig. S2). While the precipitation is not too

intense, it falls persistently over many hours, leading to high 24-h rainfall accumulation at some

locations. Thus, the local thermodynamic conditions seem to maintain for many hours and lead to

the continuous development of precipitating clouds during much of 8 December.

At 1200 UTC 9 December, a warm, moist, and unstable atmosphere is still maintained over

central Vietnam and the SCS, with $\theta_e > 335$ K (Fig. 10a and Figs. 4). However, the strong

convergence of the low-level northeasterly wind carrying the moisture in Ha Tinh and Quang Tri

provinces moved southward to Quang Tri and Quang Nam provinces (Fig. 10a). This moving

dragging along the move of the low-level moisture convergence (Figs. 10c,d). Besides, Fig. 9e

shows that the low-level uplifting motion is stronger than the previous day due to most of the strong

northeasterly wind zone blocked by the Truong Son range. Besides, the southward movement of the

northeasterly wind and moisture convergence zone also led to the southward movement of

precipitable water between the surface and 200 hPa to the coastal zone between Quang Binh and

Quang Tri provinces (Fig. 10f). As a result, the main heavy rainfall also moved southward to this

area. Moreover, these thermodynamic conditions played a role to sustain the development of

precipitating clouds on 9 December. This also coincides with observed satellite and radar data. In

detail, on this day (since 1200 UTC), satellite imageries also show some characteristics of deep

convection over the coastal area (Fig. S3), but the cloud top temperatures, in general, are not as cold

as on 8 December. Meanwhile, the lower precipitating Ns clouds cover much of the study area from
1200 UTC 9 to 0300 UTC 10 December, then gradually disintegrate (Fig. S3). These clouds kept
producing rainfall for the whole day, with the higher $C_{max}$ values (~40 dBZ) and rainfall intensity
from 1200 UTC 9 to around 0000 UTC 10 December (Fig. S4), mainly over the coastal plain and
nearby sea. After that, the rain gradually decreases in both intensity and areal coverage.

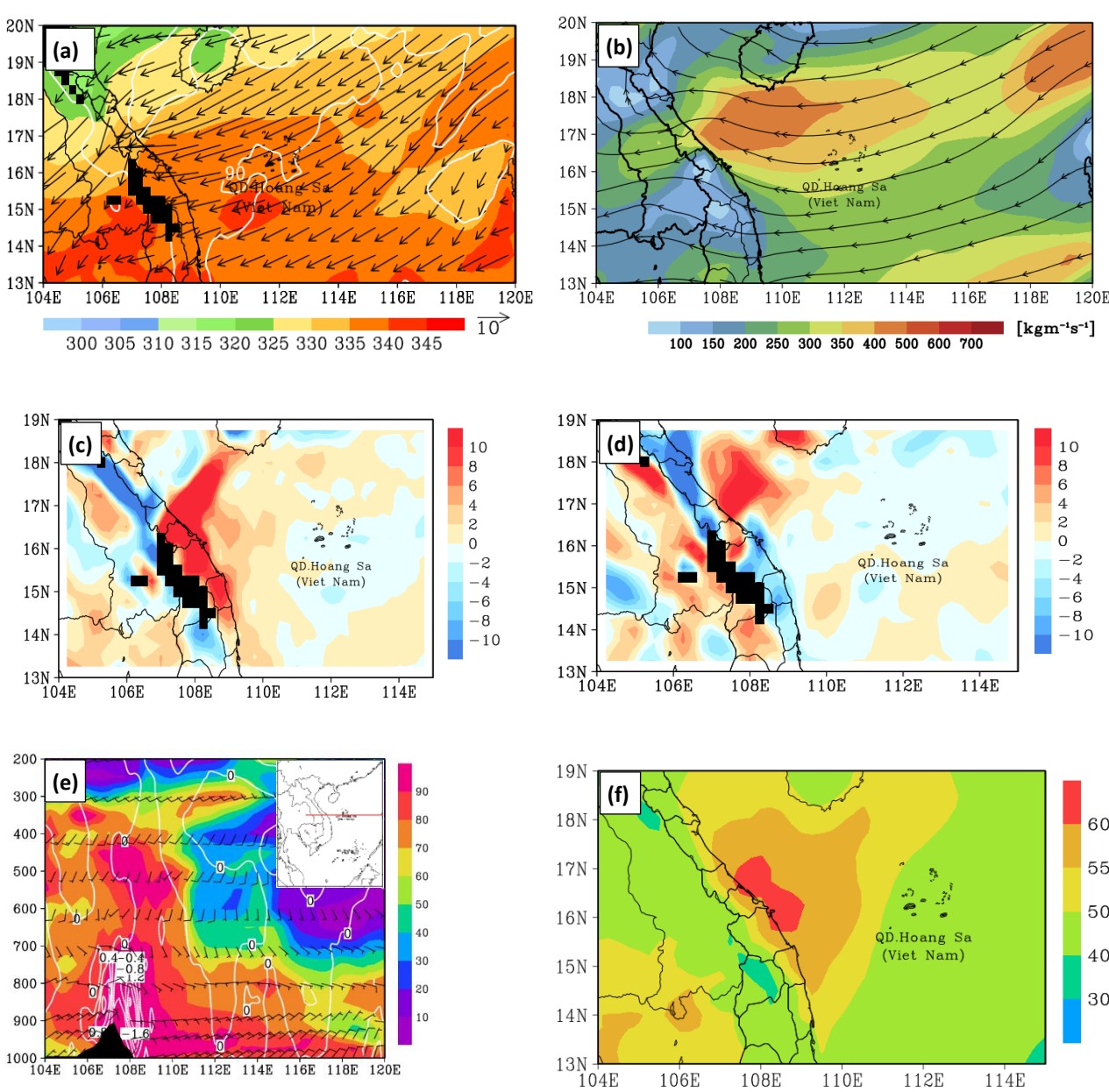

**Figure 10.** As in Fig. 9, except for 1200 UTC 9 Dec 2018.

At 1200 UTC 10 December, the atmosphere remains very moist with a precipitable water amount of 55 mm (Fig. 11d). Some of the local dynamical and thermodynamically parameters, however, are reduced from one day earlier and become not as favorable, including the velocity of northeasterly wind, the upward motion over central Vietnam (Fig. 11c), moisture flux (Fig. 11b) and precipitable water amount (Fig. 11f). Hence, the development of precipitating clouds also reduces significantly on this day and mostly exist offshore over the ocean (Fig. S5). Compared to the past two days, the development of convective cells is also reduced. Near the coast, only three convective cells developed on 10 December, one at 1400 UTC, the second at 2000 UTC, and the third one shortly after 2200 UTC. Also, moving eastward and offshore after formation, these relatively small cells spend only 1-3 h over land. In general, the environmental conditions become less favorable for developing rain clouds after 1200 UTC 10 December. Consequently, there is a significant decrease in rainfall, which occurs mainly during 1200-1600 UTC then weaken with time (Fig. S6).

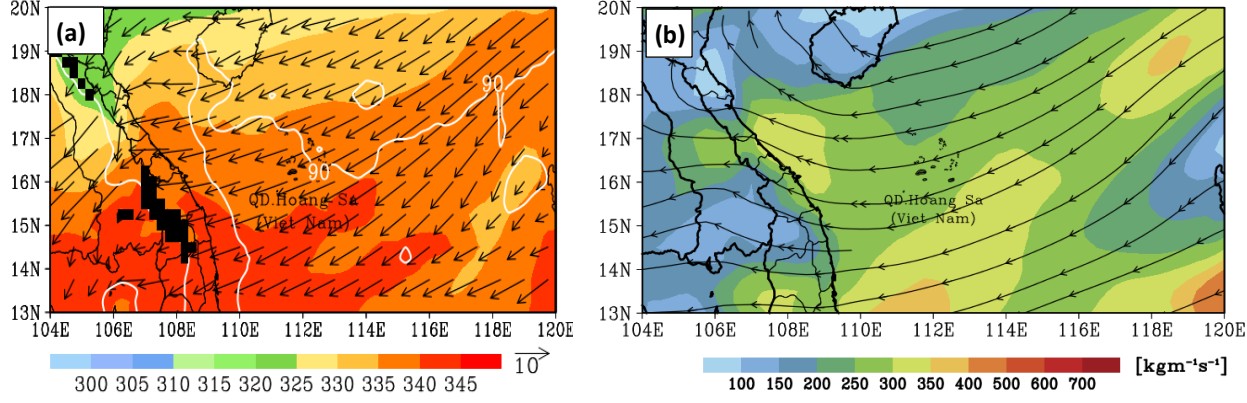

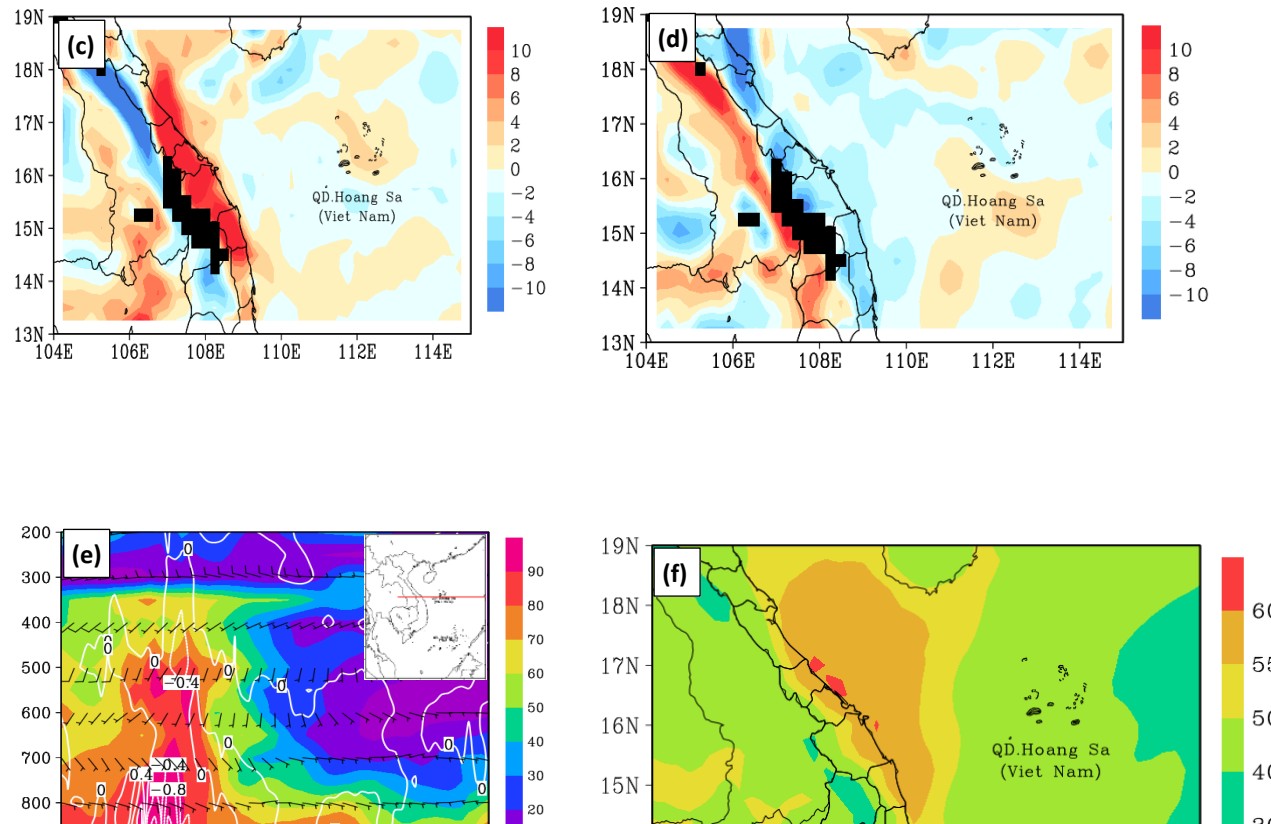

Figure 11. As in Fig. 9, except for 1200 UTC 10 Dec 2018.

**4 Model Simulation Results**

In this section, the model simulation results are used to investigate the role of topography in the development of clouds and rainfall in the D18 event, and the CReSS model is also evaluated for its ability to reproduce the event over the study area.

Figure 12 presents the daily averaged surface horizontal winds and daily rainfall in CTRL and NTRN for each of the three days from 9 to 11 December 2018. In CTRL, the model has well simulated the surface wind. As a result, the model produced a maximum 24-h rainfall of around 400 mm on 9 December (Fig. 12a), roughly comparable in magnitude to the observation (Fig. 12c). While one should bear in mind that the limited number of rain gauges have a smaller coverage area and cannot resolve the detailed distribution of rainfall (cf. Fig. 1a), the model rainfall in CTRL is

slightly more offshore north of 16° N but more inland near 16° N, thus is not as abundant along the

coast compared to the observation. In other words, model rainfall has some location errors but the

magnitude is comparable by visual inspection.

An objective and more quantitative verification of model rainfall can be provided by the threat

score (TS) computed at the rain-gauge sites, which shows that the model has high score at low

thresholds of ≤ 10 mm (per 24 h) but gradually decreases toward higher thresholds (Fig. 13a, red

curve). In particular, the TS is about 0.5 at 25-50 mm, below 0.2 above 160 mm, and about 0.1 at

350 mm. Eventually, the TS drops to zero at 500 mm, which is not too far from the observed peak

rainfall of over 500 mm (at Da Nang, cf. Fig. 1a). The bias score (BS) confirms that the model does

not produce enough rainfall over the coastal plains, as its value drops from about 1.0 at 0.05 mm to

below 0.4 at and above 250 mm. As another objective measure of overall quality of prediction, the -

Similarity skill score (SSS) is about 0.5 for 9 December. Overall, the model appears to produce too

much rainfall offshore north of 16° N and not enough rainfall along the coast, and this might be to

some extent linked to its surface wind coming more from the east-northeast, compared to northeast

in the ERA5 analysis (Figs. 12a,c), leading to somewhat different locations of low-level

convergence of wind and moisture.








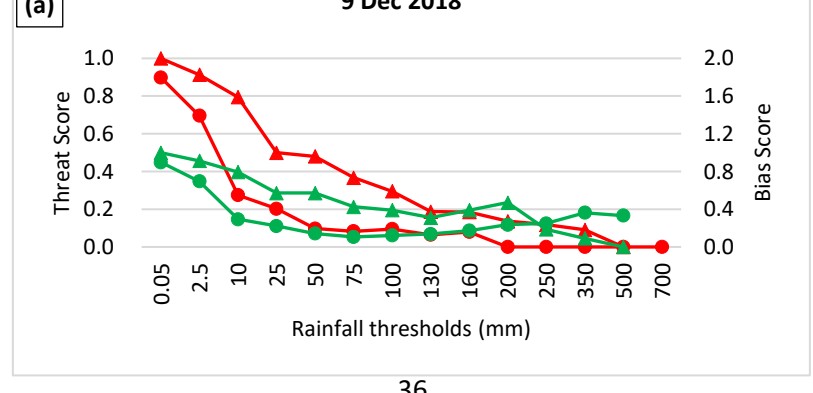

**Figure 12.** Simulated daily-mean surface horizontal wind vectors (m s$^{-1}$, reference length at right column) and 24-h accumulated rainfall (mm, color) in CTRL (left column) and NTRN (middle column), and the observed rainfall at gauge sites (OBS), overlaid with the daily-mean surface wind vectors derived from the ERA5 data (right column). From top to down are: (a-c) 9 Dec, (d-f) 10 Dec, and (g-i) 11 Dec 2018. The pink number at the lower left indicates the maximum value of 24-h rainfall.

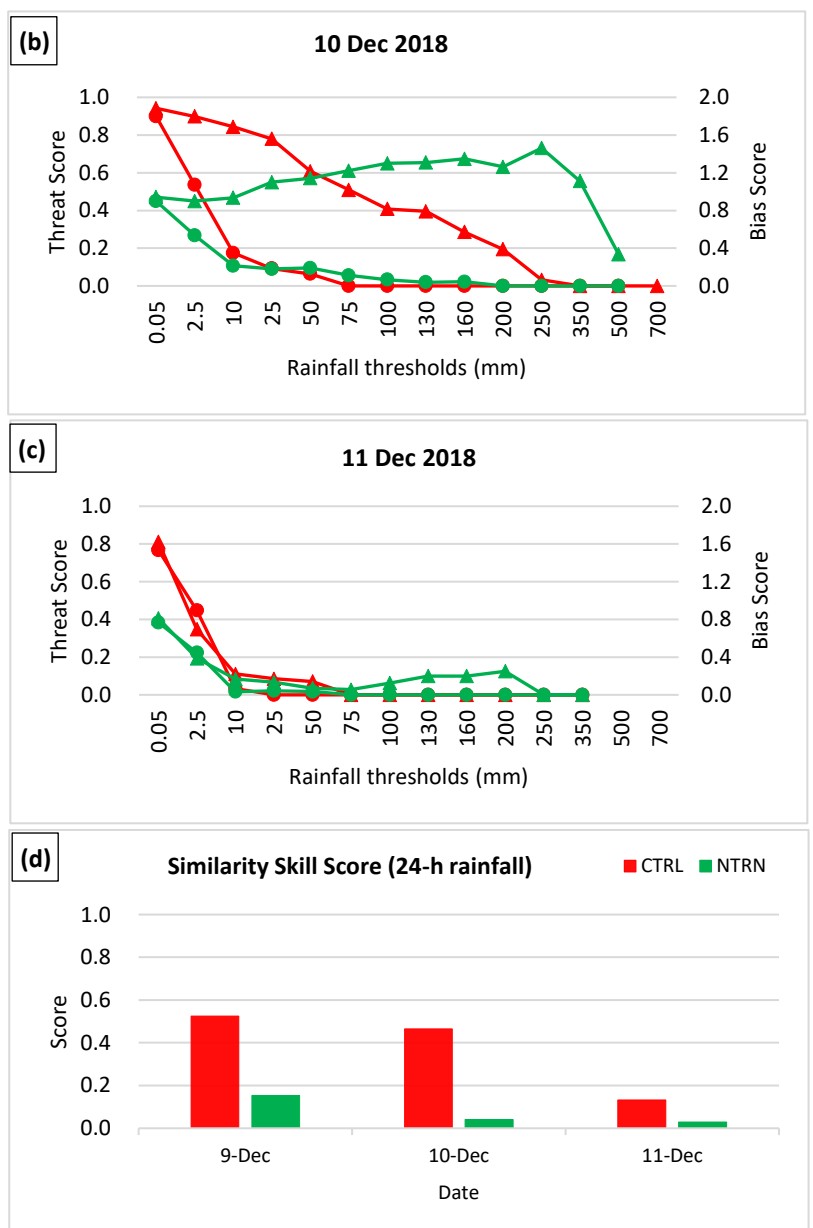

**Figure 13.** (a)-(c) The threat scores (red) and bias scores (green) of 24-h accumulated rainfall for the CTRL (curve with triangles) and NTRN (curve with dots) experiments for the three days of 9-11 Dec 2018. (d) Fractions skill scores of 24-h accumulated rainfall for the two experiments.

For 10 December, while similar differences in prevailing surface winds still exist between model simulation and ERA5 data, the model captured the southward movement of the northeasterly wind. Therefore, the model had well captured the southward movement of the main heavy rainfall. The rainfall location has improved with better agreement with the observation (Figs. 12d,f), but in

general slightly more inland and not right on the coast. Both over 600 mm, the observed and
simulated peak daily rainfall values are again comparable.  Due to the improvement in spatial
pattern, the TSs exhibit higher values than those for the previous day across low to middle
thresholds (up to 200 mm) but reduce to zero at 250 mm (Fig. 13b), while the SSS (near 0.46) is
only slightly reduced (Fig. 13d). In agreement with the better TS values, the BS remains between
0.8 and about 1.4 from low thresholds up to 350 mm, and drops to about 0.35 at 500 mm (Fig. 13b).

For 11 December, the model does not simulate well the rainfall field, as its rainfall is displaced

toward the Truong Son Range (and the border to Laos), instead of over the coastal plain as observed
(Figs. 12g,i). The spatial coverage of model rainfall is smaller and the peak amount (~200 mm) also
lower compared to the rain-gauge data, while the surface wind appears weaker than the ERA5 data
as well. While the observed peak amount became lower as the D18 event was coming to an end, the
TSs also decrease rapid with threshold, and are close to 0.1 at just 10 mm and become zero at and
above 70 mm (Fig. 13c). Consistent with the inadequate amount over land, the BSs also decrease
rapidly with thresholds, from about 0.8 at 0.05 mm to below 0.3 over 100-200 mm. For this day, the
SSS is only about 0.14 and significantly lower than the values for 9 and 10 December (Fig. 13d).
Likely also related to the weaker surface winds in the model, the less-than-ideal results of rainfall
may be also affected by the longer range of integration, at 66-90 h, for 11 December.

To test the impact of topography in the D18 event, the NTRN experiment was carried out.

Without the terrain, the model had not good simulated the surface wind. Consequences, the rainfall
as simulated by CReSS would be displaced much more inland from the coastal region for all three
days of 9-11 December (Figs. 12b,e,h), and more importantly, the pattern would no longer be
elongated and parallel to the coast, even though the peak amounts are similar to the observation.
Thus, the topography was fundamental in determining the basic rainfall area and pattern in the D18
event. With incorrect distributions, the TS values (Fig. 13, green curves) are much lower and drop
to below 0.2 at thresholds above 10-25 mm for all three days. The thresholds at which the TSs

decrease to zero are 200, 75, and 25, respectively for the three days, and much lower than those in the CTRL, especially for 9 and 10 December. The BS values in the NTRN also tend to be lower than those in the CTRL, sometimes much lower, reflecting its incorrect location and thus little rainfall at gauge sites with rainfall in reality. The SSS values are also much lower, with values near 0.16, 0.04, and 0.04 for the three days. Without the topography, the surface wind pattern near the coast and over land would be much stronger and very different, due to the lack of its blocking and uplifting effects, and also the associated thermodynamic effects.

For the D18 event as a whole, the three-day total rainfall distribution produced by the model compares quite favorably with the observation in both quantity and spatial pattern (Figs. 14a,c), with generally minor displacement errors more toward inland at around 15°-16° N. Despite these errors, the spatial distribution of rainfall in the model corresponds well to the zone of low-level moisture convergence in the ERA5 analysis (Fig. 7a). In agreement with visual assessment, the TSs of the 72-h QPFs are quite high across even heavy-rainfall thresholds: around 0.8 at 100 mm (per 72 h), close to 0.5 at 200 mm, above 0.2 at 350 mm, and 0.1 at 700 mm, with an overall SSS ≈ 0.7 (Figs. 14d,e). As shown, the rainfall fields for individual days in D18 are very different without the topography in NTRN, and the same is true for the whole event (Fig. 14b). The TSs also indicate a much lower skill in QPF, with TS below 0.2 at ≥ 50 mm (per 72 h) and TS = 0 at ≥ 350 mm, BS below 0.35 at ≥ 10 mm, and also an overall SSS of less than 0.1 (Figs. 14d,e). The results in Figs. 12 and 14 also indicate a significant wind-blocking effect by the Truong Son Range. In CTRL, the surface northeasterly winds commonly exceed 10 m s$^{-1}$ in speed over the SCS, but are reduced significantly (and even to near-zero speed) near the Annamite Range (and in Laos). On the contrary, there is no reduction in speed as the winds blow across central Vietnam in NTRN, without the blocking effect of the topography.

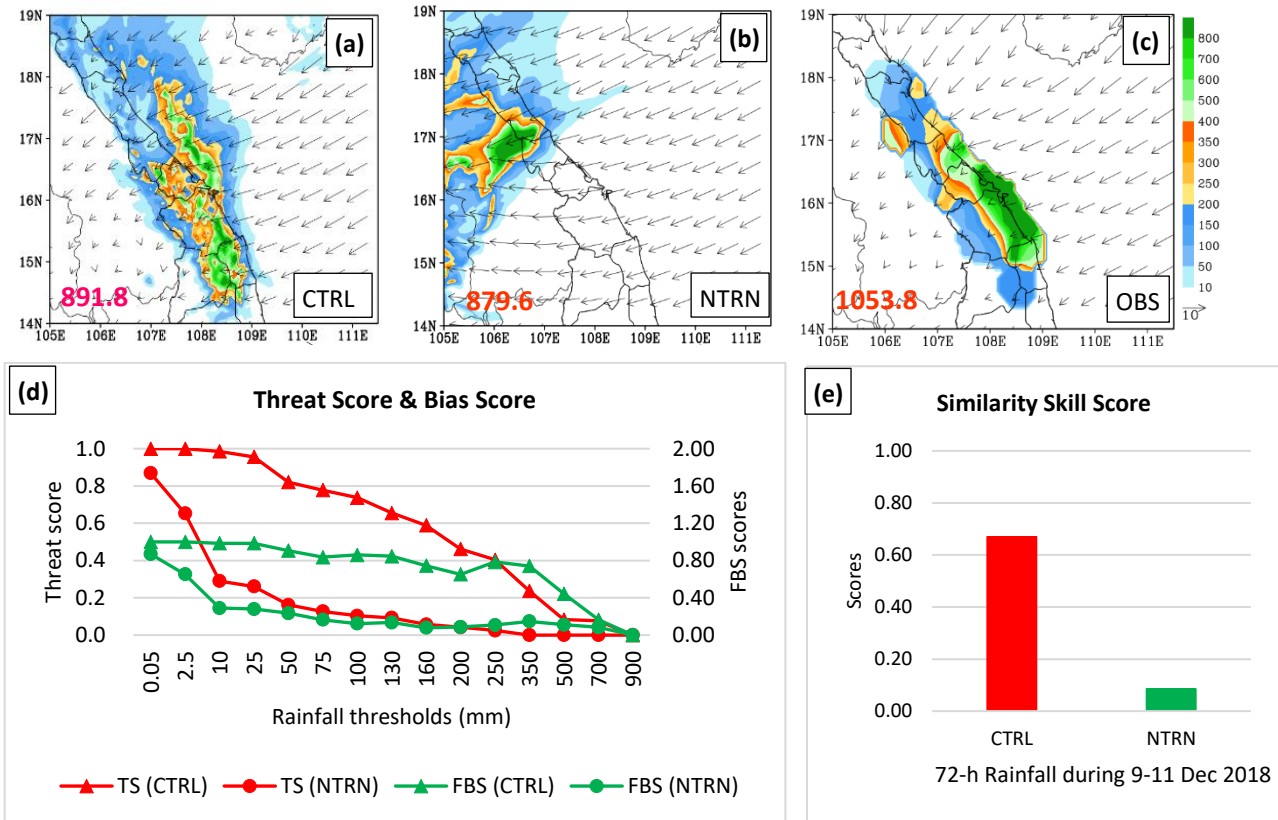

**Figure 14.** (a)-(c) As in Figs. 11a-c, except for three-day averaged surface horizontal wind vectors and 72-h accumulated rainfall over 9-11 Dec 2018. (d), (e) As in Figs. 12c,d, except for TSs and FSSs of the 72-h accumulated rainfall over 9-11 Dec 2018.

## 5 Conclusion

In this study, the extreme precipitation event that occurred on 8-12 December 2018 along the coast of central Vietnam is analyzed, and the simulation results by a CRM (the CReSS model) is evaluated. The major findings are summarized below.

Analysis on the D18 event has revealed several key factors which led to this record-breaking rainfall event: First, for all four days from 8 to 11 December, the strong northeasterly winds in the lower troposphere blew from the Yellow Sea into the SCS, and interacted with strong low-level easterly winds (below 700 hPa) over the SCS. This interaction strengthened the upstream easterly to northeasterly winds and generated strong low-level convergence, as the winds blew into central Vietnam and were blocked by the Truong Son Range, the low-level northeasterly flow reduced in

speed and led to moisture flux convergence and rising motion along the coast of Vietnam

persistently. Consequently, heavy rainfall was produced along the coast of central Vietnam. Second,

the strong easterly winds played an important role in transporting moisture from the WNP, across

the Philippines and the SCS, into central Vietnam. Third, the Truong Son Range also played an

important role in this event due to its barrier effect. Finally, the high SST of the SCS ($>27°$ C) also

acted to help replenishing the moisture in this event. This above mechanism in the D18 event is

different from those documented in previous studies. Particularly, according to previous studies, the

heavy and extreme rainfall events are usually due to the multi-interaction between the northeasterly

wind and preexisting tropical disturbance over the SCS and local topography or tropical cyclone or

impacts by ENSO or MJO. However, these factors have not appeared during the D18 event.

Therefore, we suggest that the interaction of the northeasterly and easterly winds in the moist,

unstable atmospheric and local topography can also lead to heavy precipitation events along the

central coastal plains of Vietnam. Another interesting finding of this study is that even though short

periods of heavy rainfall from deep convection also contributed, the extreme rainfall of the D18

event was mainly from the persistent rain from nimbostratus clouds (Ns) that do not possess a high

reflectivity or a very cold cloud top.

One of the features of the D18 event is that the main heavy rain band moved from the north to

south of the study area during the event. The analysis of the local thermodynamic reveals the

movement of the convergence northeasterly wind zone in the north of the study area from north to

south. This movement dragged along the movement of the convergent moisture zone. The movement

of convergent moisture zone results in precipitation water column moving from north to south.

Consequently, the main heavy rain band moved from north to south.

The evaluation of model simulation results at a grid size of 2.5 km indicates the following. In the

CTRL, the model has well simulated the surface wind as well as captured the wind convergence's

southward movement. Therefore, the CReSS model has reproduced this event's rainfall field quite

well, for both daily and three-day accumulations, but with some displacement errors. In terms of
objective verification skill scores, in particular, CReSS displays high skills at heavy-rainfall
thresholds for both daily rainfall (TS ≥ 0.1 at 200-350 mm and FSS ≈ 0.5 for 9 and 10 December)
and 72-h total (TS ≈ 0.1 at 700 mm and FSS ≈ 0.7). However, the rainfall simulation is less ideal for
11 December (TS drops to zero at thresholds ≥ 75 mm), which had less rainfall and is at a longer
range (than the previous two days). Besides, the model also captured the southward movement of the
main heavy rain band during the event, as seen in the observed data. In the sensitivity test of NTRN
where the topography is removed, the model has poorly simulated the surface wind and did not
capture the southward movement of the wind convergence zone. This led to the model produced a
different rainfall pattern not along the coast as observed (and in CTRL), thus confirming the important
role by the Truong Son Range in this event. In addition, the evaluation of simulation results also
shows that the CReSS model has well simulated the surface winds, both in their direction and
magnitude.
Generally, these results enhanced our knowledge about the mechanisms which cause the heavy
rainfall in central Vietnam, as well as explained features of the D18 event. The above result also
shows the promising capacity of the CReSS model for research and forecast of heavy rainfall in
Vietnam. In a follow-up paper, a set of high-resolution time-lagged ensemble prediction is performed
using the CReSS model, and the predictability of the D18 event will be evaluated.
**Code and data availability**
The CReSS model used in this study and its user's guide are available at the model website at
http://www.rain.hyarc.nagoyau.ac.jp/~tsuboki/cress_html/index_cress_eng.html.
**Author contribution**
Duc Van Nguyen prepared datasets, executed the model experiments, performed analysis, and
prepared the first draft of the manuscript. Chung-Chieh Wang provided the funding, guidance and
suggestions during the study, and participated in the revision of the manuscript.

**Competing interests**

The authors declare that they have no conflict of interest.
**Acknowledgement.** We thank Mr. Nguyen Tien Toan at Mid-central Regional Hydro-
Meteorological Centre, Viet Nam for kindly providing the observed rainfall and radar data, as well
as his comment. We acknowledge the free use of ECMWF ERA5 from Copernicus Climate Change
Service (C3S) Climate Data Store (CDS) https://www.ecmwf.int/en/forecasts/datasets/ reanalysis-
datasets/era5. The Vietnam Gridded Precipitation rainfall dataset is available at
http://danida.vnu.edu.vn/cpis/en/content/gridded-precipitation-data-of-vietnam.html. The TRMM
3B42 satellite data are from https://disc.gsfc.nasa.gov/datasets/TRMM_3B42_7/summary. The IR1
Himawari imagines data are from Central Weather Bureau, Taiwan at https://www.cwb.gov.tw.

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
