# Peer review of "Investigation of An Extreme Rainfall Event during 8-12 December 2018 over"

_Natural Hazards and Earth System Sciences, 2022_

## Referee Comment (RC1)

Article: "Investigation of An Extreme Rainfall Event during 1 8-12 December 2018 over Central Vietnam. Part I: Analysis and Cloud-Resolving Simulation" Research on the problems of record heavy rainfall in the central part of Vietnam is very interesting for the readers.

This rain is caused by the influence of cold air and strong high-altitude westerly wind, followed by strong easterly winds, so from the evening of December 7, it was heavy rain in Quang Tri. After that, it rained heavily, until 3 am on December 8, it rained heavily in Da Nang until about 9 am, then it rained heavily on the coast of the Quang Nam Sea. Rain mainly concentrated from the evening of December 8 to the end of December 10, and on December 11 and 12, the area still had rain but the intensity decreased.

**COMMENTS**

- After reading the article, readers have some comments as follows:

- - Part 1: The overview of the article has not mentioned much about the situation causing heavy rain and the ability to solve the problem of heavy rain caused by this weather pattern in Vietnam.

- - Part 2: It is necessary to describe more clearly the two options for removing terrain and not removing terrain in the experiment. Additional options for physics of Cress model.

- - Part 3: analyzes a lot about the weather patterns that cause rain but still does not explain the cause of rain for this period.

- - Part 4: The forecasted rainy area with the case of keeping the topography (Ctl) gives the rain center deviation from reality and also does not simulate the rain well in the Truong Son mountain range. It should be noted that in this case of heavy rain, the topography is not the main factor, as evidenced by very heavy rains at coastal stations (400-600mm/day) and less rain at stations in mountainous areas.

- - Note the activities of weather patterns such as the combination of cold air with the high easterly wind and the activity of the westerly wind channel.

**Question:**

1/ It is necessary to clarify how many hours are the rain analysis periods of the cress model? Rain spreads from the north to the south, but it shows as cumulative rain in the article, so can the model describe this phenomenon?

2/ Compare the experiment with keeping the terrain with removing the terrain to explain what?

3/ In fact, the time of heavy rain of the rain being studied is short, the center of heavy rain moves from north to south, so the total of 3 days in the article is heavy rain on a large scale, not suitable for this rain. What is the cause of the occurrence of heavy rainfall in a short period of time on a small scale in this case?

**DECISION:** MAJOR REVISION

| Principal criteria | Excellent (1) | Good (2) | Fair (3) | Poor (4) |
|---|---|---|---|---|
| **Scientific Significance:** Does the manuscript represent a substantial contribution to the understanding of natural hazards and their consequences (new concepts, ideas, methods, or data)? | | | X | |
| **Scientific Quality:** Are the scientific and/or technical approaches and the applied methods valid? Are the results discussed in an appropriate and balanced way (clarity of concepts and discussion, consideration of related work, including appropriate references)? | | | X | |
| **Presentation Quality:** Are the scientific data, results and conclusions presented in a clear, concise, and well-structured way (number and quality of figures/tables, appropriate use of technical and English language, simplicity of the language)? | | | X | |

1. Does the paper address relevant scientific and/or technical questions within the scope of NHESS?

   No

2. Does the paper present new data and/or novel concepts, ideas, tools, methods or results?

   yes

3. Are these up to international standards?

   yes

4. Are the scientific methods and assumptions valid and outlined clearly?

   No

5. Are the results sufficient to support the interpretations and the conclusions?

   No

6. Does the author reach substantial conclusions?

   No

7. Is the description of the data used, the methods used, the experiments and calculations made, and the results obtained sufficiently complete and accurate to allow their reproduction by fellow scientists (traceability of results)?

   No

8. Does the title clearly and unambiguously reflect the contents of the paper?

   Yes

9. Does the abstract provide a concise, complete and unambiguous summary of the work done and the results obtained?

   Yes

10. Are the title and the abstract pertinent, and easy to understand to a wide and diversified audience?

    Yes

11. Are mathematical formulae, symbols, abbreviations and units correctly defined and used? If the formulae, symbols or abbreviations are numerous, are there tables or appendixes listing them?

    Yes

12. Is the size, quality and readability of each figure adequate to the type and quantity of data presented?

    Yes

13. Does the author give proper credit to previous and/or related work, and does he/she indicate clearly his/her own contribution?

    Yes

14. Are the number and quality of the references appropriate?

    Yes

15. Are the references accessible by fellow scientists?

    Yes

16. Is the overall presentation well structured, clear and easy to understand by a wide and general audience?

    No

17. Is the length of the paper adequate, too long or too short?

    Too short

18. Is there any part of the paper (title, abstract, main text, formulae, symbols, figures and their captions, tables, list of references, appendixes) that needs to be clarified, reduced, added, combined, or eliminated?

    Part 2, 3 and 4

19. Is the technical language precise and understandable by fellow scientists?

    Yes

20. Is the English language of good quality, fluent, simple and easy to read and understand by a wide and diversified audience?

    yes

21. Is the amount and quality of supplementary material (if any) appropriate?

    No

---

## Author Comment (AC1)

NHESS-2022-82

Authors' Responses to Reviewer 1 (RC1, anonymous)

Date: 20 July 2022

Title: Investigation of An Extreme Rainfall Event during 8-12 December 2018 over Central Vietnam. Part I: Analysis and Cloud-Resolving Simulation

Authors: C. C. Wang and Duc V. N.

Firstly, **we thank the reviewer for spending valuable time reviewing the paper and giving us helpful comments that helped to improve the clarity of the paper.**

**COMMENTS**

**Comment 01:** Part 1: The overview of the article has not mentioned much about the situation causing heavy rain and the ability to solve the problem of heavy rain caused by this weather pattern in Vietnam.

**Reply:** Thank you for your comment. In the revision, we will try our best to add more analysis as much as we can to clarify better the situation causing heavy rainfall in central Vietnam as well as the ability to solve the problem of heavy rain caused by this weather pattern in Vietnam.

**Comment 02:** Part 2: It is necessary to describe more clearly the two options for removing terrain and not removing terrain in the experiment. Additional options for physics of Cress model.

**Reply:** Thank you for your suggestion. We have added more information to make it more clearly.

**Table 1.** The basic information of experiments.

| Domain and Basic setup | |
|---|---|
| Model domain | 3°–26°N; 98°–120°E |
| Grid dimension $(x, y, z)$ | $912 \times 900 \times 60$ |
| Grid spacing $(x, y, z)$ | 2.5 km × 2.5 km × 0.5 km* |
| Projection | Mercator |
| IC/BCs (including SST) | *NCEP GDAS/FNL Global Gridded Analyses and Forecasts (*0.25° × 0.25°, every 6 h, 26 pressure levels*)* |
| Topography (for CTRL only) | Digital elevation model by JMA at (1/120)° spatial resolution |
| Simulation length | 114 h |
| Output frequency | 1 hour |
| **Model physical setup** | |
| Cloud microphysics | Bulk cold-rain scheme (six species) |
| PBL parameterization | 1.5-order closure with prediction of turbulent kinetic energy (Deardorff, 1980; Tsuboki and Sakakibara, 2007) |
| Surface processes | Energy and momentum fluxes, shortwave and longwave radiation (Kondo, 1976; Louis et al., 1982; Segami et al., 1989) |
| Soil model | 41 levels, every 5 cm deep to 2 m |

**Comment 03:** Part 3: analyzes a lot about the weather patterns that cause rain but still does not explain the cause of rain for this period.

**Reply:** Based on the thermodynamics obtained from ERA-5, we found out some key factors that caused this extreme rainfall event. (1) The interaction between the strong northeasterly winds, blowing from the Yellow Sea into the northern South China Sea (SCS), and easterly winds over the SCS in the lower troposphere (below 700 hPa). This interaction created strong low-level convergence, as the winds continued to blow into central Vietnam against the Truong Son Range, the low-level easterly flow reduced in speed and led to moisture flux convergence and rising motion along the coast of Vietnam persistently. These low-level convergence and rising motion were strong enough to trigger

most of the convection near the shoreline, instead of over the slopes (further inland) by forced uplift of the terrain. As a consequence, heavy rainfall occurred along the coast. (2) The strong easterly wind played an important role in transporting moisture from the western North Pacific across the Philippines and the SCS into central Vietnam at low-level atmosphere while the southeasterly winds between 700 hPa and 500 hPa also play important role in complementing moisture from the SCS into central Vietnam. (3) The Truong Son Range also contributed to this event due to its barrier effect. (4) In addition to cumulonimbus, the low-level precipitating clouds such as nimbostratus clouds were also major contributors to rainfall accumulation for the whole event.

Some of our results are also consistent with the identification of Dr. Hoang Phuc Lam - Deputy Director of the National Center for Meteorological Forecasting about this event on the Communist Party of Vietnam Online Newspaper (https://dangcongsan.vn/xa-hoi/mua-lon-tai-mien-trung-la-bieu-hien-ro-ret-cua-bien-doi-khi-hau---507626.html or English version:https://scienceinfo.net/rain-and-flood-in-central-region-why-do-not-forecast-rainfall-in-each-area.html). We will point out these in the revision.

**Comment 04:** Part 4: The forecasted rainy area with the case of keeping the topography (Ctl) gives the rain center deviation from reality and also does not simulate the rain well in the Truong Son mountain range. It should be noted that in this case of heavy rain, the topography is not the main factor, as evidenced by very heavy rains at coastal stations (400-600mm/day) and less rain at stations in mountainous areas.

**Reply:** Thank you for your comment. We will do our best to better clarify the deficiency of the model in heavy rainfall locations in the revision. Besides, to explain why the heavy rainfall only concentrates on narrowing coastal plain and coastal sea. We verified many aspects of this event using multiple data sources, such as thermodynamics obtained from ERA5 (Figs. 9 -11), satellite colour-enhanced infrared imageries of blackbody cloud-top temperatures and Column-maximum radar reflectivity (dBZ) over central Vietnam for every single day (supplement data). We found that the interaction between the strong northeasterly winds, blowing from the Yellow Sea into the northern South China Sea (SCS), and easterly winds over the SCS in the lower troposphere (below 700 hPa) created strong low-level convergence, as the winds continued to blow into central Vietnam against the Truong Son Range, the low-level easterly flow reduced in speed and led to moisture flux convergence and rising motion along the coast of Vietnam persistently. These low-level convergence and rising motion were strong enough to trigger most of the convection near the shoreline, instead of over the slopes (further inland) by forced uplift of the terrain. As a consequence, heavy rainfall occurred along the coast. Furthermore, the CReSS test

without the terrain (NTRN run) also indicates that the rainfall pattern is no longer parallel to the coastline and dissimilar to the observation. Therefore, we think in D18 event the terrain played an important role to block the low-level flow and led to moisture flux convergence and rising motion (to initiate convection repeatedly). We will do our best to point these out in the revision.

**Comment 05:** Note the activities of weather patterns such as the combination of cold air with the high easterly wind and the activity of the westerly wind channel.

**Reply:** Thank you for your comment. We will do our best to better clarify these activities of weather patterns in the revision.

**Question**

**Question 01.** It is necessary to clarify how many hours are the rain analysis periods of the cress model? Rain spreads from the north to the south, but it shows as cumulative rain in the article, so can the model describe this phenomenon?

**Reply:** The satellite and radar data (Figs. S1-6) show that the precipitating clouds, including cumulonimbi (Cb) and nimbostratus (Ns), produced long-lasting rainfall for hours in the study area. While the precipitation is not too intense, it falls persistently over many hours, leading to high 24-h rainfall accumulation at some locations. Compared with the CReSS CTRL results indicate that it is somewhat similar in the spatial distribution between 24-h model rainfall and precipitable clouds as well as radar reflectivity over coastal plain and coastal sea. Furthermore, Figs. 12a,d the rainfall moved from north to south in CTRL as observed with some location errors in the main rainband. Therefore, we think the model can simulate the D18 event, however, with some location errors in peak amounts.

**Question 02.** Compare the experiment with keeping the terrain with removing the terrain to explain what?

**Reply:** Many previous studies showed that the local topography plays an important role in the formation of heavy rainfall events in central Vietnam although the local mountains are not really height (< 3000 m). Furthermore, analyses of the thermodynamics of this event also indicate that the local topography plays an important role in this event due to its barrier effect. Hence, we executed these two experiments to verify it as well as to see how the

rainfall was distributed without the terrain. The result of these two experiments showed the important role of local terrain in the formation and distribution of rainfall in this event.

**Question 03.** In fact, the time of heavy rain of the rain being studied is short, the center of heavy rain moves from north to south, so the total of 3 days in the article is heavy rain on a large scale, not suitable for this rain. What is the cause of the occurrence of heavy rainfall in a short period of time on a small scale in this case?

**Reply:** The satellite and radar data indicated that the continuous development of precipitating clouds known as nimbostratus (Ns) produced long-lasting rainfall for hours. Although the precipitation is not too intense, it falls persistently over many hours, added by short spikes of intense rainfall from deep convection (Figs. S1-S6), leading to high 24-h rainfall accumulation at some locations.

---

## Author Comment (AC2)

NHESS-2022-82

Authors' Responses to Reviewer 2 (RC2, anonymous)

Date: 01 Aug 2022

Title: Investigation of An Extreme Rainfall Event during 8-12 December 2018 over Central Vietnam. Part I: Analysis and Cloud-Resolving Simulation

Authors: C. C. Wang and Duc V. N.

First of all, **we thank the reviewer for the valuable comments that have significantly improved the clarity and highlighted important points of the paper**

**Major comments:**

**Comment 01**. I think the manuscript needs an extensive and thorough reorganisation to improve the presentation of the authors' idea.

**Reply:** In the revision, we re-organized our manuscript by removing the parts that give similar information to other parts, adding more information to clarify our idea in paragraphs that are not clear, and changing some sentence positions to make the manuscript suitable, as suggested.

**Comment 02**. The motivation of the sensitivity study on the role of local terrain on the D18 event is unclear as the role of local terrain in heavy rainfall in central Vietnam seems to be well understood (Lines 76-79; 83-85).

**Reply:** Many previous studies showed that the local topography plays an important role in the formation of heavy rainfall events in central Vietnam although the local mountains are not really height (< 3000 m). Besides, analyses of the thermodynamics of this event also indicate that the local topography plays an important role in this event due to its barrier effect. Furthermore, these tests can also help clarify the reason why the heaviest rainfall was along the coast and not over the mountain slopes in D18. Hence, we executed these two experiments to verify it as well as to see how the rainfall was distributed without the terrain. The result of these two experiments showed the important role of local terrain in the formation and distribution of rainfall in this event. We will add more information as

well as describe more detail to highlight our motivation for the sensitivity study on the role of local terrain in the D18 event.

**Comment 03**. The motivation of using CReSS is not well presented in the introduction. Yet, the motivation of using CReSS can be found in later sections (Lines 171-172; 480-482).

**Reply:** In the revision, we rearranged our manuscript by moving lines 171-172 to the "introduction" part and added more information to make it more clearly as follows:

"In recent decades, the Cloud-Resolving Storm Simulator (CReSS) has been widely known due to its good performance in quantitative precipitation forecasts. This model has been applied to study tropical cyclones, heavy to extreme rainfall events, and many other convective systems in Japan and Taiwan (e.g., Ohigashi and Tsuboki, 2007; Yamada *et al.*, 2007; Akter and Tsuboki, 2010, 2012; Wang *et al.*, 2015). Furthermore, the CReSS model has been used to perform routine high-resolution forecasts at the National Taiwan Normal University (NTNU) and provided to the TTFRI as a forecast member since 2010. Hence, this study employed the CReSS model to simulate the D18 event and evaluated its performance."

**Comment 04.** In the conclusion, the authors stated that "according to previous studies, the heavy and extreme rainfall events are usually due to the multi-interaction between the northeasterly wind and preexisting tropical disturbance over the SCS and local topography or tropical cyclone or impacts by ENSO or MJO. However, these factors have not appeared during the D18 event". I found this conclusion quite problematic:

- o Although it should be obvious that the D18 event was not related to preexisting tropical disturbance/cyclones (see Figures in Supplement), the authors should have pointed this out in the analysis.

- o The potential impact of ENSO and/or MJO on the D18 event was not analysed in this study, thus I am not sure how the authors drew such a conclusion.

**Reply:** You are right. We will add the following analysis about these factors in the revision to clarify it as your suggestion.

*Regarding tropical disturbance/cyclones:*

"Figure 4 and the figures in the Supplement also indicate that no tropical cyclone existed during the D18 event. Therefore, tropical cyclones or the combined effect of cold surges originating from northern China and tropical depressions that have been mentioned as

one of the patterns that cause heavy rainfall in central Vietnam is not the mechanism of the D18 event". This analysis will be added into Line 229

*Regarding the potential impact of ENSO and/or MJO on the D18 event:*

"Besides investigating the synoptic-scale atmospheric conditions above, this study also verifies the impact of intraseasonal oscillations in the tropical atmosphere on the D18 event. To be more specific, figure 8a reveals that the MJO in Western Pacific was not active in early December 2018 as well as during the D18 event. Figure 8b indicates that the last three months of 2018 are a fairly weak El Niño phase. In addition, previous studies showed that central Vietnam had less rainfall in the El Niño years. Therefore, MJO and ENSO are not the cause and have no impact on the D18 event

[Figure]

Figure 8. (a) The Madden-Julian Oscillation (MJO) location and the strength through 8 different areas along the equator around the globe. Labelled dots for each day. Red line is for October, Green line is for November, Blue line is for December. Source: Commonwealth of Australia 2019, Bureau of Meteorology. (b) The Oceanic Niño Index (ONI) of the Niño 3.4 region (5° N-5° S, 120°-170° W) for 2018".

This information will be added right after Line 353.

**Comment 05.** I think the authors could have compared the cause of extreme rainfall events, which are not related to tropical distributions/cyclones, and the cause of the D18 event. This can truly pin down the key factors that led to the D18 event.

**Reply:** This is a good suggestion. We will do our best to compare D18 event with non-TC related events in the past to identify the key differences in the revision, as suggested.

**Comment 06**. Some analyses appear to be irrelevant to the overall objectives of this study. For example, the use of TRMM and related analysis could be excluded from this study.

**Reply:** Due to the limitation of the observation station network, we only have the observation stations inland. Therefore, we used the satellite data to support our analysis of the distribution of the main rainfall over the coastal sea, as shown in the figure. 1e and fig. 1d. In the revision, we will add more information to clarify the purpose of using TRMM data.

**Comment 07**. Some sections appear to be repetitive, for example, Section 3.2 and part of Section 3.3 give very similar information.

**Reply:** In section 3.2, we computed and analysed the three days average atmospheric conditions. This allows us to see the main factors that govern the weather condition in the study area during the D18 event. In section 3.3, we analyse the thermodynamics of every single day to see how thermodynamics evolves day by day. This can help us understand why the rainfall and its spatial distribution are very different day by day. However, because the rainfall and its distribution are very different day by day. Therefore, we have decided to remove subsection 3.2 to avoid repetition, as suggested.

**Minor comments:**

**Lines 12-13**: Remove "and its simulation … is evaluated."

**Reply:** You are right. This mentioned part seems to give very similar information to the sentence located on lines 16-17. So, this part was deleted.

**Line 15**: what "easterly wind" is the author referring to? What region of "high sea surface temperature" is the author referring to?

**Reply:** Easterly wind refers to the low-level winds that blow from the east-to-west prevailing direction and originate from the northwest pacific. It can be seen clearly in Fig. 4b. The high sea surface temperature region refers to a part of the Northwest Pacific Ocean and South China Sea where the sea surface temperature is higher than 27° C.

Based on your confusion, we added more information to each factor to make it more clearly as follows: "easterly wind" to "low-level easterly wind blow to central Vietnam from the northwest pacific ocean" and "high sea surface temperature" to "high sea surface temperature over North West Pacific ocean and South China Sea."

**Lines 17-18**: This statement is too general, perhaps add some details of the results.

**Reply:** You are right. We re-written the sentence and added more specific information as follow: "The cloud-resolving model can simulate the extreme rainfall event both in quantitative rainfall and its spatial distribution, however with some location errors in peak amounts".

**Line 34**: Remove "at high resolution".

**Reply:** We think that "high resolution" is remarkable information. So we replaced "at high resolution" by "at a grid size of 2.5 km" to keep the relevant information. The new sentence will be "The Cloud-Resolving Storm Simulator (CReSS) was employed to simulate this record- breaking event at a grid size of 2.5 km, and the overall rainfall …".

**Lines 38-40**: I think the author forgot to mention a key finding here as the spatial distribution of the rainfall is also different in the NTRN experiment.

**Reply:** You are right. We had updated the information in the new sentence: "In the sensitivity test without the terrain, the model not only did not generate nearly as much rainfall for this event but also did not capture the spatial distribution of the rainfall".

**Line 44**: I suppose the words "disasters" and "hazards" have the same meaning in this manuscript. It might be a good idea to stick with one of them for consistency.

**Reply:** We agree with you. We had decided to use the word "disasters" for consistency.

**Lines 54-55**: Change "… according to climate change and sea-level rise scenarios for Vietnam…" to "… according to a publication by the Ministry of Natural Resources and Environment of Vietnam (Tran et al. 2016) …"

**Reply:** This is a good suggestion. We had changed "… according to climate change and sea-level rise scenarios for Vietnam…" to "according to a publication by the Ministry of Natural Resources and Environment of Vietnam (Tran et al. 2016) regarding climate change and sea-level rise scenarios, extreme rainfall events will increase in both their frequency and intensity in the future".

**Figure 2**: This figure could be merged with Figure 1.

**Reply:** Thanks for your suggestion, we had merged Figure 2 with Figure 1 and make new Figure 1 as follow, as your suggestion.

[Figure]

[Figure]

**Figure 1.** (a) observed 24 h accumulated rainfall (mm, color dots, 1200 – 1200 UTC) and topography (m, shaded) for 9 Dec. Vertical colorbar for rainfall, and horizontal colorbar for topography. (b) As in (a), but for 10 Dec. (c) As in (a), but for 11 Dec.  (d) As in (a), but for 72 h accumulated rainfall during 1200 UTC 8–1200 UTC 11 Dec. (e) 72 h accumulated rainfall obtained by TRMM estimate. The pink dot marks the location of Da Nang station. (f) Mean annual rainfall distribution (mm) in Vietnam from 1980 to 2010, obtained from the Vietnam Gridded Precipitation (VnGP) data, and the study area of central Vietnam (red box).

**Lines 72-75**: This paragraph should be merged with the next paragraph.

**Reply:** Corrected.

**Lines 72-85**: This should appear before the paragraph starting in line 58. the description of the local topography (lines 62-64) should be included in this paragraph.

**Reply:** Thanks for your suggestion. We will make changes in the revision, as suggested.

**Lines 86-102**: Currently this paragraph is disjointed from the previous paragraphs. The authors can add a sentence to connect this paragraph with the previous paragraphs.

**Reply:** You are right. We used the linking word "Furthermore" to connect these two paragraphs.

**Lines 104-106**: The authors would need to offer more evidence to support these statements. One idea is to plot the time series of annual maximum 72-h accumulated rainfall from 1980-2018 (depending on the available data) and highlight the maximum 72-h accumulated rainfall of the D18 event.

**Reply:** Thank you for your suggestion. In the revision, we cited a statement of Dr Lam - Deputy Director of the National Center for Hydro-Meteorological Forecasting that is "however, according to Dr. Hoang Phuc Lam – National Center for Hydro-Meteorological Forecasting, it can be said that this extreme event has never happened in the past because the observed rainfall at some places in the Central region has surpassed the record according to the statistics of rainfall at the end of the main rainy season (Communist Party of Vietnam Online Newspaper)", as suggested.

**Line 108**: At this point, I am not sure why local terrain is an interesting factor to investigate. As the authors have pointed out, many factors including the local topography can cause heavy rainfall in central Vietnam (Lines 76-79). Perhaps there is a very good reason behind it, but it is not well communicated at this point.

**Reply:** Although the local topography is just one of many factors that can cause heavy rainfall in central Vietnam. However, the local topography usually plays an important role in unusually rainfall events that occur in the main rainy season (from late fall to early winter). The reason is that central Vietnam is strongly affected by low-level northeasterly winds that originate from northern China, and low-level easterly winds blow from the Northwest Pacific Ocean. Therefore, when these winds blow to central Vietnam, they will be prevented there due to the local topography. This can be seen in fig 7b. Hence, we would like to investigate whether the local topography contributes to the D18 event as it usually does in previous heavy rainfall events. We will clarify this point in the revision.

**Line 109**: Remove "or high-resolution".

**Reply:** Removed.

**Line 119**: Change "This dataset is …" to "The NCEP GDAS/FNL Global Gridded Analyses and Forecasts is …".

**Reply:** Changed.

**Lines 128-129**: Why ERA5 is used for this purpose instead of NCEP GDAS/FNL? Conversely, why ERA5 is not used as IC/BCs for the CRM simulation?

**Reply:**

*Regarding why ERA5 is used for this purpose instead of NCEP GDAS/FNL*

In fact, both ERA5 and NCEP GDAS/FNL can serve our study. Therefore, when we start the study, we choose an independent dataset (ERA5) to delineate the synoptic weather patterns during the D18 event.

*Regarding why ERA5 is not used as IC/BCs for the CRM simulation:*

Actually, we had a CRM simulation using the ERA5. However, the result is not good as the CRM simulation using NCEP GDAS/FNL data as IC/BCs. Therefore, we have chosen the simulated results using NCEP GDAS/FNL data as IC/BCs to present in our manuscript.

[Figure]

**Lines 141-146**: Why TRMM is used? What is the added value of using this dataset in this study?

**Reply:** Due to the limitation of the observation station network, we only have the observation stations inland. Therefore, we used the satellite data to see rainfall distribution over the coastal sea, as shown in the figure. 1e and fig. 1d.

**Line 172**: The meaning of "large computers" is not clear.

**Reply:** You are right. We had changed this word to "parallel computers"

**Table 1**: Not sure about the meaning of "Real" for the data source of topography input.

**Reply:** To avoid confusion, we had changed "Real" to "Digital elevation model"

**Line 193**: Remove "=".

**Reply:** Removed.

**Line 197:** Change "correct negative" to "true negative" or "correct rejection".

**Reply:** Changed.

**Line 199**: Remove "(where CN is not used)".

**Reply:** Removed.

**Table 2:** The definition of the "worst score" for BS is not very clear. Based on the formulation of BS, BS = 0 if FA = 0 and H = 0, i.e. the model always predictor negative. In a way, the worst-case scenario could be H = 0 and FA = All Negative, i.e. the model is predicting the opposite result 100% of the time, i.e. BS = FA/M but this is not equal to N.

**Reply:** Based on the Bias Score formula (BS) =(H+FA)/(H+M). It can be seen that the worst BS is FA/M, and its worst possible value for over-prediction is M = 1 (minimum) and FA = N – 1 (maximum), so that BS = N – 1, as your pointed out. In the manuscript, we wrote BS = N as a close approximation to that (if N is large). However, to better clarify, in the revision, we had changed the "worst score" from "0 or N" to "0 or N – 1".

**Lines 211-212:** "As introduced earlier, … central Vietnam." This sentence seems to be redundant.

**Reply:** We used this sentence to start our description of the D18 event more smoothly. However, I removed this sentence based on your comment.

**Line 215:** The term "windward side" might be a bit confusing as the wind field is only shown in Figure 4. Perhaps change it to eastern (or north-eastern) side?

**Reply:** Changed to "eastern side".

**Line 216:** It might be useful to indicate the Quang Nam province in Figure 1.

**Reply:** After merging the Figure 2 with the figure 1a. We had indicated the Quang Nam province in Figure 1a. It can be seen in the reply to minor comment #Figure 2 above.

**Line 221:** "In this subsection, … are analyzed." This sentence seems to be redundant.

**Reply:** We agree with you, we had removed this sentence.

**Lines 221-283:** I think this part can be streamlined in a more concise manner.

**Reply:** This part is also checked and improved in fluency, as suggested.

**Line 290:** Perhaps rename the subtitle as "The local thermodynamic conditions prior the D18 event"

**Reply:** After we re-organized our manuscript structure, we removed subsection 3.2, and now line 290 (or subsection 3.3) was named "**Synoptic conditions**".

**Line 291:** Remove the sentence "In this part, the … analyzed."

**Reply:** Removed.

**Line 325:** "atmospheric"?

**Reply:** We "corrected" it to atmosphere.

**Figures 9, 10, 11:** It might be better to merge all the graphs together and to remove the (a) panels as they more or less carry the same information as the (b) panels. The resultant 3x3 panel plot would be a better presentation to the evolution of the environment of each day. Readers would find it easier to understand the changes.

**Reply:** Thanks for your suggestion. We will make changes in the revision, as suggested.

**Figure 12:** The date convention is not consistent with the earlier Figures, e.g. Figure 1.

**Reply:** Corrected.

**Line 389:** Is the difference in prevailing surface winds between CTRL and OBS linked to the use of two different data sets (NCEP GDAS/FNL for CTRL and ERA5 for OBS)?

**Reply:** Sorry for making you confused about it. We made it consistent with the simulated results by using NCEP GDAS/FNL surface winds for OBS.

**Lines 423-442, Figure 14:** I am not sure about the added value of this part. It seems to be repeating the previous section.

**Reply:** This part evaluates the model's performance in the total 3-day rainfall over the entire event, since each day had somewhat different rainfall characteristics. To be more specific, the rainfall and its spatial distribution as well as between the CTRL and NTRN experiments are very different day by day. Therefore, the model will present different skills on a specific day as well as a specific experiment. Furthermore, when we consider the three days of the D18 event as a whole. The rainfall will be larger, and its spatial distribution also closer to reality. Therefore, when considering the D18 as a whole, we can see the model's predictability at higher rainfall thresholds.

**Line 470:** I think the authors downplay the impact of displacement errors in their simulations. These errors would have significant impact from the disaster risk reduction perspective. In a way, being able to correctly forecast the location of extreme rainfall occurrence is as important, if not more important, as being able to forecast the amount of extreme rainfall.

**Reply:** We agree with you. we will point out this the deficiency of the model in heavy rainfall location more detail in the revision. However, as we know, it is almost impossible to simulation deep convection (stochastic in nature) at the right location, and the model is not perfect. Therefore, if the model can capture the magnitude and used to study the cause of the D18 event, it is already quite good and worthwhile.

**Reference:**

Chen, B., Yu, W., Wang, W., Zhang, Z., Dai, W. (2021). A global assessment of precipitable water vapor derived from GNSS zenith tropospheric delays with ERA5, NCEP FNL, and NCEP GFS products. Earth and Space Science, 8, e2021EA001796. https://doi.org/10.1029/2021EA001796

Mun, J., Jeon, W., Lee, H. W. (2020). Impact of Different Meteorological Initializations on WRF Simulation During the KORUS-AQ Campaign. J. Environ. Sci. Int., 29, 33-44. https://doi.org/10.5322/JESI.2020.29.1.33

Zhu, X.-M., Song, X.-N., Leng, P., Zhou, F.-C., Gao, L., G., D. (2022). Performances of six reanalysis profile products in the atmospheric correction of passive microwave data for estimating land surface temperature under cloudy-sky conditions. International Journal of Digital Earth, 15:1, 296-322, DOI: 10.1080/17538947.2022.2025463

---

## Author Response (AR1)

NHESS-2022-82

Authors' Responses to Reviewers (anonymous)

Date: 14 Oct 2022

Title: Investigation of An Extreme Rainfall Event during 8-12 December 2018 over Central Vietnam. Part I: Analysis and Cloud-Resolving Simulation

Authors: C. C. Wang and Duc V. N.

First of all, **we thank the reviewer for the valuable comments that have significantly improved the clarity and highlighted important points of the paper**

**Reviewer 1**

**Comment 02:** Part 2: It is necessary to describe more clearly the two options for removing terrain and not removing terrain in the experiment. Additional options for physics of Cress model.

**Respond:** We have added more information to make it more clearly as below. The information can be seen in table 1 in the revised version.

**Table 1.** The basic information of experiments.

| Domain and Basic setup | |
|---|---|
| Model domain | 3°–26°N; 98°–120°E |
| Grid dimension (*x*, *y*, *z*) | 912 × 900 × 60 |
| Grid spacing (*x*, *y*, *z*) | 2.5 km × 2.5 km × 0.5 km* |
| Projection | Mercator |
| IC/BCs (including SST) | *NCEP GDAS/FNL Global Gridded Analyses and Forecasts* (0.25° × 0.25°, every 6 h, 26 pressure levels*)* |
| Topography (for CTRL only) | Digital elevation model by JMA at (1/120)° spatial resolution |
| Simulation length | 114 h |
| Output frequency | 1 hour |

| Model physical setup | |
| --- | --- |
| Cloud microphysics | Bulk cold-rain scheme (six species) |
| PBL parameterization | 1.5-order closure with prediction of turbulent kinetic energy (Deardorff, 1980; Tsuboki and Sakakibara, 2007) |
| Surface processes | Energy and momentum fluxes, shortwave and longwave radiation (Kondo, 1976; Louis et al., 1982; Segami et al., 1989) |
| Soil model | 41 levels, every 5 cm deep to 2 m |

**Comment 03:** Part 3: analyzes a lot about the weather patterns that cause rain but still does not explain the cause of rain for this period.

**Respond:** Based on the thermodynamics obtained from ERA-5, we found out some key factors that caused this extreme rainfall event. (1) The interaction between the strong northeasterly winds, blowing from the Yellow Sea into the northern South China Sea (SCS), and easterly winds over the SCS in the lower troposphere (below 700 hPa). This interaction created strong low-level convergence, as the winds continued to blow into central Vietnam against the Truong Son Range, the low-level easterly flow reduced in speed and led to moisture flux convergence and rising motion along the coast of Vietnam persistently. These low-level convergence and rising motion were strong enough to trigger most of the convection near the shoreline, instead of over the slopes (further inland) by forced uplift of the terrain. As a consequence, heavy rainfall occurred along the coast. (2) The strong easterly wind played an important role in transporting moisture from the western North Pacific across the Philippines and the SCS into central Vietnam at low-level atmosphere while the southeasterly winds between 700 hPa and 500 hPa also play important role in complementing moisture from the SCS into central Vietnam. (3) The Truong Son Range also contributed to this event due to its barrier effect. (4) In addition to cumulonimbus, the low-level precipitating clouds such as nimbostratus clouds were also major contributors to rainfall accumulation for the whole event.

Some of our results are also consistent with the identification of Dr. Hoang Phuc Lam - Deputy Director of the National Center for Meteorological Forecasting about this event on the Communist Party of Vietnam Online Newspaper (https://dangcongsan.vn/xa-hoi/mua-lon-tai-mien-trung-la-bieu-hien-ro-ret-cua-bien-doi-khi-hau---507626.html    or    English

version:).  We pointed this out in the revised version.

**Comment 04:** Part 4: The forecasted rainy area with the case of keeping the topography (Ctl) gives the rain center deviation from reality and also does not simulate the rain well in the Truong Son Range. It should be noted that in this case of heavy rain, the topography is not the main factor, as evidenced by very heavy rains at coastal stations (400-600mm/day) and less rain at stations in mountainous areas.

**Respond:** We added more information to clarify the deficiency of the model in heavy rainfall locations in the revision. Besides, to explain why the heavy rainfall only concentrates on narrowing coastal plain and coastal sea. We verified many aspects of this event using multiple data sources, such as thermodynamics obtained from ERA5 (Figs. 9 - 11), satellite colour-enhanced infrared imageries of blackbody cloud-top temperatures and Column-maximum radar reflectivity (dBZ) over central Vietnam for every single day (supplement data).  We found that the interaction between the strong northeasterly winds, blowing from the Yellow Sea into the northern South China Sea (SCS), and easterly winds over the SCS in the lower troposphere (below 700 hPa) created strong low-level convergence, as the winds continued to blow into central Vietnam against the Truong Son Range, the low-level easterly flow reduced in speed and led to moisture flux convergence and rising motion along the coast of Vietnam persistently. These low-level convergence and rising motion were strong enough to trigger most of the convection near the shoreline, instead of over the slopes (further inland) by forced uplift of the terrain. As a consequence, heavy rainfall occurred along the coast. Furthermore, the CReSS test without the terrain (NTRN run) also indicates that the rainfall pattern is no longer parallel to the coastline and dissimilar to the observation. Therefore, we think in D18 event the terrain played an important role to block the low-level flow and led to moisture flux convergence and rising motion (initiate convection repeatedly). We had pointed these out in the revision.

**Comment 05:** Note the activities of weather patterns such as the combination of cold air with the high easterly wind and the activity of the westerly wind channel.

**Respond:** We added more information to clarify these activities of weather patterns in the revision.

**Reviewer 2**

**Major comments:**

**Comment 01**. I think the manuscript needs an extensive and thorough reorganisation to improve the presentation of the authors' idea.

**Respond:** In the revision:

- we reorganised our manuscript by rearranging the paragraphs to make the manuscript more suitable and adding more information to clarify our idea in paragraphs that were not clear.
- We decided to keep section 3.2 because we believed that the time-averaged of selected variables would help us easily to highlight the main patterns that caused this event.
- Especially, we added more data and reanalyzed section 3.3 to clarify the southward movement of the main heavy rain band during the D18 event.

**Comment 02**. The motivation of the sensitivity study on the role of local terrain on the D18 event is unclear as the role of local terrain in heavy rainfall in central Vietnam seems to be well understood (Lines 76-79; 83-85).

**Respond:** Many previous studies showed that the local topography plays an important role in the formation of heavy rainfall events in central Vietnam although the local mountains are not really height (< 3000 m). Besides, analyses of the thermodynamics of this event also indicate that the local topography plays an important role in this event due to its barrier effect. Furthermore, these tests can also help clarify the reason why the heaviest rainfall was along the coast and not over the mountain slopes in D18. Hence, we executed these two experiments to verify it as well as to see how the rainfall was distributed without the terrain. The result of these two experiments showed the important role of local terrain in the formation and distribution of rainfall in this event. We had added more information (line 130-132) to highlight our motivation for the sensitivity study on the role of local terrain in the D18 event.

**Comment 03**. The motivation of using CReSS is not well presented in the introduction. Yet, the motivation of using CReSS can be found in later sections (Lines 171-172; 480-482).

**Respond:** In the revision, we rearranged our manuscript by moving lines 171-172 to the "introduction" part and added more information to make it more clearly as follows:

"In recent decades, the Cloud-Resolving Storm Simulator (CReSS) has been widely known due to its good performance in quantitative precipitation forecasts. This model has been applied to study tropical cyclones, heavy to extreme rainfall events, and many other convective systems in Japan and Taiwan (e.g., Ohigashi and Tsuboki, 2007; Yamada *et al.*, 2007; Akter and Tsuboki, 2010, 2012; Wang *et al.*, 2015). Furthermore, the CReSS model has been used to perform routine high-resolution forecasts at the National Taiwan Normal University (NTNU) and provided to the TTFRI as a forecast member since 2010. Hence, this study employed the CReSS model to simulate the D18 event and evaluated its performance." This information can be found on lines 117-124.

**Comment 04.** In the conclusion, the authors stated that "according to previous studies, the heavy and extreme rainfall events are usually due to the multi-interaction between the northeasterly wind and preexisting tropical disturbance over the SCS and local topography or tropical cyclone or impacts by ENSO or MJO. However, these factors have not appeared during the D18 event". I found this conclusion quite problematic:

- Although it should be obvious that the D18 event was not related to preexisting tropical disturbance/cyclones (see Figures in Supplement), the authors should have pointed this out in the analysis.

- The potential impact of ENSO and/or MJO on the D18 event was not analysed in this study, thus I am not sure how the authors drew such a conclusion.

**Respond:** We had added the following analysis about these factors in the revision to clarify it as your suggestion.

*Regarding tropical disturbance/cyclones:*

"Besides, Figure 3 also indicates that there was no existence of any tropical cyclone during the D18 event. Therefore, tropical cyclones or the combined effect of cold surges originating from northern China and tropical depressions that have been mentioned as one of the patterns that cause heavy rainfall in central Vietnam is not the mechanism of the D18 event.". This information can be found on lines 274 -277.

*Regarding the potential impact of ENSO and/or MJO on the D18 event:*

"Besides investigating the synoptic-scale atmospheric conditions above, this study also verifies the impact of intraseasonal oscillations in the tropical atmosphere on the D18 event. To be more specific, figure 8a reveals that the MJO in Western Pacific was not

active in early December 2018 as well as during the D18 event. Figure 8b indicates that the last three months of 2018 are a fairly weak El Niño phase. In addition, previous studies showed that central Vietnam had less rainfall in the El Niño years. Therefore, MJO and ENSO are not the cause and have no impact on the D18 event

[Figure]

Figure 8. (a) The Madden-Julian Oscillation (MJO) location and the strength through 8 different areas along the equator around the globe. Labelled dots for each day. Red line is for October, Green line is for November, Blue line is for December. Source: Commonwealth of Australia 2019, Bureau of Meteorology. (b) The Oceanic Niño Index (ONI) of the Niño 3.4 region (5° N-5° S, 120°-170° W) for 2018".

This information can be found on Lines 194 -201 and right after Line 338.

**Comment 05.** I think the authors could have compared the cause of extreme rainfall events, which are not related to tropical distributions/cyclones, and the cause of the D18 event. This can truly pin down the key factors that led to the D18 event.

**Respond:** All the causes of extreme rainfall events in the past were related to tropical distributions/cyclones/ENSO, MJO, … which are mentioned in section 1. So, we focused on the analysis of the D18 event to clarify the cause.

**Comment 06**. Some analyses appear to be irrelevant to the overall objectives of this study. For example, the use of TRMM and related analysis could be excluded from this study.

**Respond:** Due to the limitation of the observation station network, we only have the observation stations inland. Therefore, we used the satellite data to support our analysis of the distribution of the main rainfall over the coastal sea, as shown in the figure. 1e and fig.

1d. In the revision, we had added more information to clarify the purpose of using TRMM data. This information can be found on lines 173 -175.

**Comment 07**. Some sections appear to be repetitive, for example, Section 3.2 and part of Section 3.3 give very similar information.

**Respond:** In section 3.2, we computed and analysed the three days time-average atmospheric conditions. This allows us to see the main factors that govern the weather condition in the study area during the D18 event. To avoid repetition, in section 3.3, we focused on analysing the changes in local thermodynamics every single day to see the answer to the question of why the main heavy rain band occurred in the coastal zone and why they moved southward during the event.

**Minor comments:**

**Lines 12-13**: Remov "and its simulation … is evaluated."

**Respond:** This part was deleted.

**Line 15**: what "easterly wind" is the author referring to? What region of "high sea surface temperature" is the author referring to?

**Respond:** Easterly wind refers to the low-level winds that blow from the east-to-west prevailing direction and originate from the northwest pacific. It can be seen clearly in Fig. 4b. The high sea surface temperature region refers to a part of the Northwest Pacific Ocean and South China Sea where the sea surface temperature is higher than 27° C.

we had added more information to each factor to make it more clearly as follows: "easterly wind" to "low-level easterly wind blow to central Vietnam from the northwest pacific ocean" and "high sea surface temperature" to "high sea surface temperature over North West Pacific ocean and South China Sea.". this can be seen on lines 14 – 16.

**Lines 17-18**: This statement is too general, perhaps add some details of the results.

**Respond:** We re-written the sentence and added more specific information as follow: "The cloud-resolving model can simulate the extreme rainfall event both in quantitative rainfall and its spatial distribution, however with some location errors in peak amounts".

**Line 34**: Remove "at high resolution".

**Respond:** We think that "high resolution" is remarkable information. So we replaced "at high resolution" by "at a grid size of 2.5 km" to keep the relevant information. The new sentence will be "The Cloud-Resolving Storm Simulator (CReSS) was employed to simulate this record- breaking event at a grid size of 2.5 km, and the evaluated results …". This change can be seen on line 39

**Lines 38-40**: I think the author forgot to mention a key finding here as the spatial distribution of the rainfall is also different in the NTRN experiment.

**Respond:** We had updated the information in the new sentence: "In the sensitivity test without the terrain, the model not only did not generate nearly as much rainfall for this event but also did not capture the spatial distribution of the rainfall". This information can be found on lines 45 -47.

**Line 44**: I suppose the words "disasters" and "hazards" have the same meaning in this manuscript. It might be a good idea to stick with one of them for consistency.

**Respond:** We had decided to use the word "disasters" for consistency. The This information can be found on line 53.

**Lines 54-55**: Change "… according to climate change and sea-level rise scenarios for Vietnam…" to "… according to a publication by the Ministry of Natural Resources and Environment of Vietnam (Tran et al. 2016) …"

**Respond:** This is a good suggestion. We had changed "… according to climate change and sea-level rise scenarios for Vietnam…" to "according to a publication by the Ministry of Natural Resources and Environment of Vietnam (Tran et al. 2016) regarding climate change and sea-level rise scenarios, extreme rainfall events will increase in both their frequency and intensity in the future". This information can be found on lines 61 -64

**Figure 2**: This figure could be merged with Figure 1.

**Respond:** We had merged Figure 2 with Figure 1 and make new Figure 1 as follow. This information can be found the revised version.

[Figure]

[Figure]

**Figure 1.** (a) observed 24 h accumulated rainfall (mm, color dots, 1200 – 1200 UTC) and topography (m, shaded) for 9 Dec. Vertical colorbar for rainfall, and horizontal colorbar for topography. (b) As in (a), but for 10 Dec. (c) As in (a), but for 11 Dec. (d) As in (a), but for 72 h accumulated rainfall during 1200 UTC 8–1200 UTC 11 Dec. (e) 72 h accumulated rainfall obtained by TRMM estimate. The pink dot marks the location of Da Nang station. (f) Mean annual rainfall distribution (mm) in Vietnam from 1980 to 2010, obtained from the Vietnam Gridded Precipitation (VnGP) data, and the study area of central Vietnam (red box).

**Lines 72-75**: This paragraph should be merged with the next paragraph.

**Respond:** Corrected. This changed can be seen on lines 66 -81

**Lines 72-85**: This should appear before the paragraph starting in line 58. the description of the local topography (lines 62-64) should be included in this paragraph.

**Respond:** We had made changed in the revision. These changes can be seen on lines 66 - 88.

**Lines 86-102**: Currently this paragraph is disjointed from the previous paragraphs. The authors can add a sentence to connect this paragraph with the previous paragraphs.

**Respond:** We used the linking word "Furthermore" to connect these two paragraphs. This change can be seen on line 100

**Lines 104-106**: The authors would need to offer more evidence to support these statements. One idea is to plot the time series of annual maximum 72-h accumulated rainfall from 1980-2018 (depending on the available data) and highlight the maximum 72-h accumulated rainfall of the D18 event.

**Respond:** In the revision, we cited a statement of Dr Lam - Deputy Director of the National Center for Hydro-Meteorological Forecasting that is "however, according to Dr. Hoang Phuc Lam – National Center for Hydro-Meteorological Forecasting, it can be said that this extreme event has never happened in the past because the observed rainfall at some places in the Central region has surpassed the record according to the statistics of rainfall at the end of the main rainy season (Communist Party of Vietnam Online Newspaper)". This information can be found on lines 126 -130.

**Line 108**: At this point, I am not sure why local terrain is an interesting factor to investigate. As the authors have pointed out, many factors including the local topography can cause heavy rainfall in central Vietnam (Lines 76-79). Perhaps there is a very good reason behind it, but it is not well communicated at this point.

**Respond:** Although the local topography is just one of many factors that can cause heavy rainfall in central Vietnam. However, the local topography usually plays an important role in unusually rainfall events that occur in the main rainy season (from late fall to early winter). The reason is that central Vietnam is strongly affected by low-level northeasterly winds that originate from northern China, and low-level easterly winds blow from the Northwest Pacific Ocean. Therefore, when these winds blow to central Vietnam, they will be prevented there due to the local topography. This can be seen in fig 6b. Hence, we would like to investigate whether the local topography contributes to the D18 event as it usually does in previous heavy rainfall events. We had also added more information to clarify this point in the revision. This information can be found on lines 130- 133.

**Line 109**: Remove "or high-resolution".

**Respond:** Removed. This change can be seen on line 136.

**Line 119**: Change "This dataset is …" to "The NCEP GDAS/FNL Global Gridded Analyses and Forecasts is …".

**Respond:** Changed. This change can be seen on line 146.

**Lines 128-129**: Why ERA5 is used for this purpose instead of NCEP GDAS/FNL? Conversely, why ERA5 is not used as IC/BCs for the CRM simulation?

**Respond:**

*Regarding why ERA5 is used for this purpose instead of NCEP GDAS/FNL*

In fact, both ERA5 and NCEP GDAS/FNL can serve our study. Therefore, when we start the study, we choose an independent dataset (ERA5) to delineate the synoptic weather patterns during the D18 event.

*Regarding why ERA5 is not used as IC/BCs for the CRM simulation:*

Actually, we had a CRM simulation using the ERA5. However, the result is not good as the CRM simulation using NCEP GDAS/FNL data as IC/BCs. Therefore, we have chosen the simulated results using NCEP GDAS/FNL data as IC/BCs to present in our manuscript.

[Figure]

**Lines 141-146**: Why TRMM is used? What is the added value of using this dataset in this study?

**Respond:** Due to the limitation of the observation station network, we only have the observation stations inland. Therefore, we used the satellite data to see rainfall distribution

over the coastal sea, as shown in the figure. 1e and fig. 1d. we also add more information to clarify our purpose. This information can be seen on lines 173 -175.

**Line 172**: The meaning of "large computers" is not clear.

**Respond:** We rewritten this section 2.2 and removed this word. This change can be seen in section 2.2 in the revised version.

**Table 1**: Not sure about the meaning of "Real" for the data source of topography input.

**Respond:** we changed "Real" to "Digital elevation model". This information can be found in table 1.

**Line 193**: Remove "=".

**Respond:** Removed. The change can be seen on line 239

**Line 197:** Change "correct negative" to "true negative" or "correct rejection".

**Respond:** Changed. The change can be seen on line 243

**Line 199**: Remove "(where CN is not used)".

**Respond:** Removed. The change can be seen on line 245

**Table 2:** The definition of the "worst score" for BS is not very clear. Based on the formulation of BS, BS = 0 if FA = 0 and H = 0, i.e. the model always predictor negative. In a way, the worst-case scenario could be H = 0 and FA = All Negative, i.e. the model is predicting the opposite result 100% of the time, i.e. BS = FA/M but this is not equal to N.

**Respond:** Based on the Bias Score formula (BS) =(H+FA)/(H+M). It can be seen that the worst BS is FA/M, and its worst possible value for over-prediction is M = 1 (minimum) and FA = N – 1 (maximum), so that BS = N – 1, as your pointed out. In the manuscript, we wrote BS = N as a close approximation to that (if N is large). However, to better clarify, in the revision, we had changed the "worst score" from "0 or N" to "0 or N – 1". This change can be seen in table 2

**Lines 211-212:** "As introduced earlier, … central Vietnam." This sentence seems to be redundant.

**Respond:** We removed this sentence. This change can be seen on line 258.

**Line 215:** The term "windward side" might be a bit confusing as the wind field is only shown in Figure 4. Perhaps change it to eastern (or north-eastern) side?

**Respond:** Changed to "eastern side". This change can be seen on line 261.

**Line 216:** It might be useful to indicate the Quang Nam province in Figure 1.

**Respond:** After merging the Figure 2 with the figure 1a. We have indicated the Quang Nam province in Figure 1a. This information can be found in Figure 1a.

**Line 221:** "In this subsection, … are analyzed." This sentence seems to be redundant.

**Respond:** We removed this sentence. The change can be seen on line 267.

**Lines 221-283:** I think this part can be streamlined in a more concise manner.

**Respond:** This part is also checked and improved in fluency, as suggested.

**Line 290:** Perhaps rename the subtitle as "The local thermodynamic conditions prior the D18 event"

**Respond:** It was renamed "The local thermodynamic conditions prior the D18 event "

**Line 291:** Remove the sentence "In this part, the … analyzed."

**Respond:** Removed. This change can be seen on line 351.

**Line 325:** "atmospheric"?

**Respond:** We "corrected" it to atmosphere. This change can be seen on line 391.

**Figures 9, 10, 11:** It might be better to merge all the graphs together and to remove the (a) panels as they more or less carry the same information as the (b) panels. The resultant 3x3 panel plot would be a better presentation to the evolution of the environment of each day. Readers would find it easier to understand the changes.

**Respond:** To better understand the southward movement of the main heavy rain band during the D18 event. We added data to these figures and replot the existing picture by

zooming in on central Vietnam in the revision. This change can be seen in the revised version.

**Figure 12:** The date convention is not consistent with the earlier Figures, e.g. Figure 1.

**Reply:** Corrected. This change can be seen in the revised version.

**Line 389:** Is the difference in prevailing surface winds between CTRL and OBS linked to the use of two different data sets (NCEP GDAS/FNL for CTRL and ERA5 for OBS)?

**Reply:** We made it consistent with the simulated results by using NCEP GDAS/FNL surface winds for OBS. This change can be seen in the revised version.

**Lines 423-442, Figure 14:** I am not sure about the added value of this part. It seems to be repeating the previous section.

**Respond:** This part evaluates the model's performance in the total 3-day rainfall over the entire event, since each day had somewhat different rainfall characteristics. To be more specific, the rainfall and its spatial distribution as well as between the CTRL and NTRN experiments are very different day by day. Therefore, the model will present different skills on a specific day as well as a specific experiment. Furthermore, when we consider the three days of the D18 event as a whole. The rainfall will be larger, and its spatial distribution also closer to reality. Therefore, when considering the D18 as a whole, we can see the model's predictability at higher rainfall thresholds. So we had decided to keep this section.

**Line 470:** I think the authors downplay the impact of displacement errors in their simulations. These errors would have significant impact from the disaster risk reduction perspective. In a way, being able to correctly forecast the location of extreme rainfall occurrence is as important, if not more important, as being able to forecast the amount of extreme rainfall.

**Respond:** We had added more analyzed to point out this the deficiency of the model in heavy rainfall location more detail in the revision. The change can be seen in the revised version.